

# Deep learning based automatic grounding line delineation in DInSAR interferograms

Sindhu Ramanath Tarekere[1], Lukas Krieger[1], Dana Floricioiu[1], and Konrad Heidler[2]

[1]Remote Sensing Technology Institute, German Aerospace Center, Oberpfaffenhofen Germany
[2]Data Science in Earth Observation, Technical University of Munich, Germany

**Correspondence:** Sindhu Ramanath Tarekere (Sindhu.RamanathTarekere@dlr.de), Dana Floricioiu (Dana.Floricioiu@dlr.de)

**Abstract.** The regular and robust mapping of grounding lines is essential for various applications related to the mass balance of marine ice sheets and glaciers, especially in Antarctica and Greenland. With Differential Interferometric Synthetic Aperture Radar (DInSAR) interferograms, it is possible to accurately capture the tide-induced bending of the ice shelf at a continent-wide scale and a temporal resolution of a few days. While current processing chains typically automatically generate differential
interferograms, grounding lines are still primarily identified and delineated on the interferograms by a human operator. This method is time-consuming and inefficient, considering the volume of data from current and future SAR missions. We developed a pipeline that utilizes the Holistically-Nested Edge Detection (HED) neural network to delineate DInSAR interferograms automatically. We trained HED in a supervised manner using 421 manually annotated grounding lines for outlet glaciers and ice shelves on the Antarctic Ice Sheet. We also assessed the contribution of non-interferometric features like elevation, ice
velocity and differential tide levels towards the delineation task. Our best-performing network generated grounding lines with a median distance of 186 m from the manual delineations. Additionally, we applied the network to generate grounding lines for undelineated interferograms, demonstrating the network's generalization capabilities and potential to generate high-resolution temporal and spatial mappings.

## 1   Introduction

While there is little doubt in the amount of ice mass loss from the Antarctic Ice Sheet (AIS) (4890 [4140–5640] Gt) and Greenland Ice Sheet (2670 [1800-3540] Gt) in the last three decades (Otosaka et al., 2023; Fox-Kemper et al., 2021), there is still considerable uncertainty in the projected sea level rise from ice sheet evolution models, particularly for the AIS (Seroussi et al., 2020; Robel et al., 2019; Pattyn and Morlighem, 2020; Aschwanden et al., 2021). One of the reasons for this is the poorly understood dynamic processes that occur at ice-ocean boundaries, arising from limited observations of the grounding line (GL)
(Rignot, 2023). The location of grounding lines, where grounded ice masses transition to floating ice shelves (Weertman, 1974), serve as a gate when computing the mass balance of ice sheets with the mass budget method (Rignot and Thomas, 2002; Rignot et al., 2008). The grounding line is also where land ice first interacts with the ocean. Ice shelves melt at their bases as they come into contact with warm circumpolar deep water, leading to ice shelf thinning and a consequent GL retreat (Rignot and



Jacobs, 2002; Depoorter et al., 2013). Therefore, the position and migration patterns of GLs are indicators of ice sheet stability
(Schoof, 2007), which makes it essential to regularly map their location.

There are two main challenges when detecting the grounding line: its sub-glacial location and the short-term migration due to the tidal flexure of the ice shelf. For these reasons, existing methods detect different features serving as proxies for the grounding line (Brunt et al., 2011) as illustrated in Fig. 1a. In-situ GL detection from terrestrial (Jacobel et al., 1994; Catania et al., 2010; MacGREGOR et al., 2011) and airborne (Uratsuka et al., 1996) ice-penetrating radar aim at deriving the
true grounding line $G$. Surface slope measurements using tiltmeters (Stephenson and Doake, 1979; Stephenson, 1984; Smith, 1991), static Global Positioning Systems (GPS) (Riedel et al., 1999) and kinematic GPS (Vaughan, 1994), on the other hand, measure the hinge line location $F$. Although these techniques are necessary to validate the observations from remote sensing methods, they are limited in spatial and temporal coverage.

Elevation profiles derived from radar altimetry (Dawson and Bamber, 2017) and laser altimetry (Fricker et al., 2009; Brunt
et al., 2011; Li et al., 2020) capture pointwise, in addition to $F$, the seaward limit of the tidal flexure, $H$, and the break in slope, $I_b$ (Fig. 1a). $F$ is also identified from the radar line of sight displacement fields derived from Differential Range Offset tracking (DROT) of SAR intensity images (Marsh et al., 2013; Christianson et al., 2016; Joughin et al., 2016). The above-stated methods and other static techniques are detailed in Friedl et al. (2020).

In the phase of Differential Interferometric SAR (DInSAR) interferograms, the landward limit of ice flexure can be accurately
mapped as a continuous time-stamped delineation. Although this $G$ proxy detected by DInSAR lies a few hundred metres seaward of the real limit of tidal flexure $F$, it is considered to be a good approximation of the true grounding line (Rignot et al., 2011; Friedl et al., 2020). A DInSAR interferogram is computed as the difference between two interferograms formed from three or more repeat pass SAR acquisitions. If the assumption of constant ice velocity within the temporal baseline of the interferograms holds, the resulting differential interferogram contains detectable phase changes from the tidal flexure at the
ice sheet-ice shelf boundary. The zone of ice shelf flexure is visible as a dense fringe belt in the DInSAR phase. Typically, the landward-most fringe of the fringe belt, which corresponds to $G$, is visually identified and manually digitized as the grounding line (Fig. 1b). A less common approach is to unwrap the DInSAR phase and fit the resulting elevation data to a 1D elastic beam model, which relates the hinge line location to the vertical displacement of ice (Rignot, 1996). This technique is limited to slow-moving glaciers as the coherence of the DInSAR phase for fast-flowing ice streams is often not high enough to produce
reliable elevation estimates from phase unwrapping (Mouginot et al., 2019).

Given the growing volume of SAR acquisitions suited to detect the grounded line, replacing the labour-intensive manual grounding line delineation with scalable and automatic algorithms becomes necessary. In this study, we present a method that is based on existing automatic delineation techniques (Section 2) and is an improvement of our initial work (Ramanath Tarekere et al., 2023; Ramanath Tarekere, 2022). Our proposed automation builds on the heuristic line tracing technique by training a
convolutional neural network (CNN) to segment grounding line pixels directly on the DInSAR phase. Our pipeline handles the dataset compilation for training the CNN, applies the trained neural network and processes the network-generated delineations to obtain grounding line geometries. In order to inform the neural network about physical processes and the geometric config-





uration in the flexure zone, we also used topographical, meteorological and ice flow data and investigated their impact on the grounding line (GL) delineation outcome.

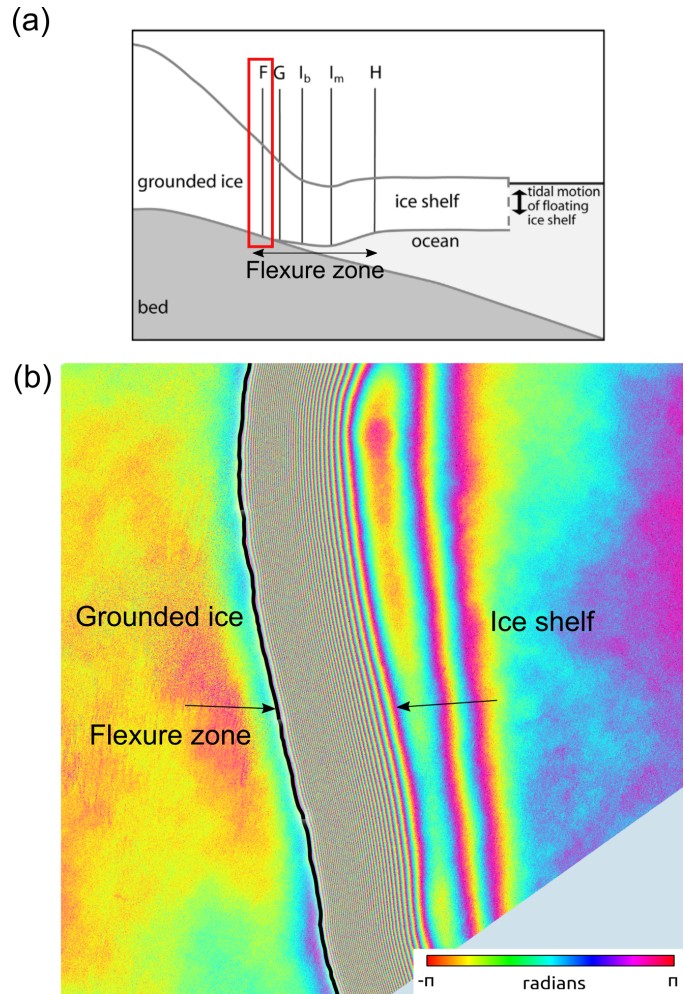

**Figure 1.** (a) The cross-section of an ice shelf, showing the features in the flexure zone: $F$ (hinge line) is the landward limit of the ice flexure due to tides, $G$ is the true grounding line, $I_b$ is the break in slope, $I_m$ is the local elevation minimum and $H$ is the seaward limit of ice flexure where the ice shelf reaches hydrostatic equilibrium. Adapted from Fricker et al. (2009). (b) DInSAR interferogram formed from two TerraSAR-X interferograms with 11 day temporal baseline. The black line shows the manually delineated GL (point $F$ in a).

## 2   Related work

Algorithms for the automated delineation of grounding lines in remotely sensed data are relatively few, likely due to the inherent complexities in tracing intricate GL geometries and the scarcity of suitable, noise-free data sources. Most of the existing GL



datasets have been generated through manual digitization of optical images (Bindschadler et al., 2011; Scambos et al., 2007) or DInSAR interferograms (Rignot et al., 2016; Groh, 2021).

Nonetheless, recent research endeavours have explored deep learning (DL) and classical model inversion techniques for automatic GL detection. Parizzi (2020) introduced a method to identify the GL within individual interferograms, circumventing the need for phase unwrapping and the generation of double-difference interferograms. The phase gradients of an interferogram are computed by estimating fringe frequencies locally within a sliding window. Subsequently, the gradient profile across the grounding zone, aligned with the fringe direction, is fitted to a one-dimensional elastic beam equation which models the tidal

deformation of the ice shelf. The grounding line's position is identified through the inversion of the fitted model. However, this method requires an a priori estimate of the grounding zone's location to constrain the potential phase gradient profiles, rendering it more semi-automatic. Additionally, the choice of the window size for fringe frequency estimation is interferogram-specific and necessitates manual adjustment.

The algorithm developed by Li et al. (2020) extracts several grounding zone features from ICESat-2 elevation anomalies

by computing the slope characteristics of the mean absolute elevation anomaly and its second derivative curves. A final visual validation is still necessary to filter out observations with a large across-track slope, as it could result in inaccurate GL positions. They used this methodology to generate an Antarctic-wide dataset comprising of 21,346 $F$, 18,149 $H$, and 36,765 $I_b$ point locations using ICESat-2 laser altimetry repeat tracks between 30 March 2019 and 30 September 2020 (Li et al., 2022).

Mohajerani et al. (2021) is the only study to date that applies deep learning for delineating the grounding line in double

difference interferograms. They used a CNN based on the DeepLabV3+ architecture (Chen et al., 2018) with parallel atrous convolutional layers and trained it on the real and imaginary components of 252 Sentinel-1 DInSAR interferograms covering the Getz Ice Shelf. They delineated grounding lines for the rest of AIS on 6 and 12 day repeat pass Sentinel-1 acquisitions in 2018. They reported a mean deviation of 232 m between the network and manual digitizations and a median absolute deviation of 101 m.

Conversely, the automatic detection of glacial calving fronts (CFs), which represent the boundaries between ice masses (grounded ice or ice shelf) and their surrounding environment (open water, sea ice, calved icebergs), has seen substantial advancement. Unlike grounding lines, calving fronts are surface features visible in optical and SAR backscatter imagery. The spectral differences between the glacier and non-glacier regions (Kääb et al., 2014) and their different backscattering response in the microwave domain make the visual identification of CFs in these modalities easier than mapping GLs in DInSAR

interferograms. As a result, numerous manually annotated datasets encompassing temporally and spatially diverse data from various satellite-based sensors are readily available and summarized e.g. in Gourmelon et al. (2022). This abundance of data has facilitated the development of classical image processing methods and, more recently, deep learning-based solutions for calving front delineation. From a computer vision perspective, GL and CF delineation are edge detection problems. In both cases, a significant class imbalance exists in the desired (GL or CF) and background pixels within any given image. Therefore,

we provide a detailed outline of developments in automatic CF delineation in Appendix A.



## 3  Dataset

### 3.1  The AIS_cci grounding line location

We used manual GL delineations from the grounding line location product of ESA's Antarctic Ice Sheet climate change initia-
tive project (hereafter AIS_cci GLL) as ground truth labels for training the neural network and to validate its performance. The
interferometric processing was carried out with a pipeline developed at the Remote Sensing Technology Institute in the Ger-
man Aerospace Center (DLR), within the AIS_cci project (Muir, 2020). The GL digitizations are available as LineStrings in
an ESRI shapefile, including metadata about acquisition conditions, tide levels and atmospheric pressure. A complete product
description can be found in Groh (2021). For this study, we used the delineations of 421 interferograms formed from Sentinel-
1, ERS-1/2 and TerraSAR-X acquisitions. Table 1 provides details regarding the temporal coverage, temporal baselines, and
the number of double difference interferograms generated from the acquisitions of the missions mentioned above. The centre
map in Fig. 2 shows the spatial extent of the AIS_cci GLLs.

**Table 1.** Overview of the satellite acquisitions of the interferograms used in the AIS_cci GLL product. IW - Interferometric Wide Swath,
SM - Stripmap. The last column shows the number of double difference interferograms used in this study.

| Satellite | Temporal extent [years] | Repeat cycle [days] | Imaging mode | No. of DInSAR interferograms |
|---|---|---|---|---|
| Sentinel 1 A/B | 2014 - 2021 | 6,12 | IW | 198 |
| ERS 1/2 | 1992 - 1996 | 1,3 | SM | 123 |
| TerraSAR-X | 2012 - 2018 | 11 | SM | 100 |

### 3.2  Training features stack

To train our neural network, we assembled a stack comprising a total of eight interferometric and non-interferometric features
which were derived from various datasets. Table 2 summarizes the feature attributes and Figure 2 shows an example of the set
of features for one interferogram.

### 3.2.1  Interferometric features

We generated the interferometric features from the double-difference wrapped phases utilized in the AIS_cci GLL production
pipeline. These include the real and imaginary components of the double difference interferograms, the wrapped phase and
pseudo coherence (enclosed in the blue polygon in Fig. 2) The pseudo coherence emerged due to resampling the wrapped
phase images while preserving cyclic phase variations ranging from $-\pi$ to $\pi$. Details of this process are given in Appendix B.
The pseudo coherence carries information about phase stability, yielding higher values in regions with lower fringe frequency.
However, unlike interferometric coherence, pseudo coherence is not an objective measure of phase quality. Decorrelated pixels



and those demonstrating good interferometric coherence with high fringe frequency are both assigned to low pseudo coherence values ($< 0.4$).

### 3.2.2 TanDEM-X PolarDEM

We used the 90-meter resolution TanDEM-X PolarDEM of Antarctica (Huber, 2020), derived from the global TanDEM-X Digital Elevation Model (Wessel, 2016) as the surface elevation feature. The TanDEM-X PolarDEM was compiled by averaging two complete coverages of bistatic acquisitions of the TanDEM-X mission conducted between April 2013 - November 2013 and April 2014 - October 2014. Additional acquisitions between July 2016 - September 2017 were used to fill gaps. Comprehensive details regarding the generation, calibration, and validation methods of TanDEM-X PolarDEM are given in Wessel et al. (2021).

### 3.2.3 Ice velocity from Sentinel-1

The ice velocity, generated by consortium partner ENVEO IT as a part of the AIS_cci project (available at http://cryoportal. enveo.at), is a 3D product containing the easting, northing and vertical components. Azimuth and line-of-sight (LOS) velocities, derived from offset tracking of SAR backscatter images (Nagler et al., 2015), were projected onto the Reference Elevation Model of Antarctica (REMA) Digital Elevation Model (Howat et al., 2019). The resulting velocity map consists of a multi-year average of maps computed from repeat-pass Sentinel-1 SAR data acquired between 2014-2021 and covering the margins of the continent up to about 75 °S. In our feature stack, we incorporated only the northing and easting components.

### 3.2.4 Ocean tide levels from Circum-Antarctic Tidal Simulation (CATS2008)

We computed the differential tidal state that is captured in the DInSAR by combining the individual tide levels derived from the CATS2008 tide model (Padman et al., 2008) in the same order as the interferograms used to create the double difference. The CATS2008 model integrates data from various sources, including tide gauges, GPS measurements, TOPEX/Poseidon radar altimetry data, ICESat-derived grounding line locations, and MOA grounding lines. While the model is gridded at 4 km, we obtained pixel-wise interpolated tide levels through Python APIs in the pyTMD module developed by Alley et al. (2017). We compensated for the change in sea surface height due to the inverse barometer effect (Padman et al., 2003) by using the daily air pressure values from the National Centers for Environmental Prediction and the National Center for Atmospheric Research Reanalysis (NCEP/NCA, provided by the NOAA PSL, Boulder, Colorado, USA, from their website at https://psl.noaa.gov). The final corrected tidal amplitudes were included in our training feature stack.





**Table 2.** Attributes of the input features used to train the neural network. No temporal coverage is specified for CATS2008 because the model assimilates several measurements across various time periods and provides the tidal amplitude for the required epoch.

| Feature | Dataset | Projection | Resolution [m] | Temporal coverage [years] |
|---|---|---|---|---|
| Phase, pseudo coherence, real and imaginary components of interferograms | AIS_cci GLL (Groh, 2021) | EPSG:4326 | S1A/B: ~17 ERS 1/2: ~18 TSX: ~5 | S1 A/B: 2014 - 2021 ERS 1/2: 1992 - 1999 TSX: 2011 - 2018 |
| DEM | TanDEM-X PolarDEM (Huber, 2020) | EPSG:3031 | 90 | April 2013 - Oct 2014 July 2016 - Sept 2017 |
| Ice velocity (northing and easting) | ENVEO IT (Wuite, 2020) | EPSG:3031 | 200 | 2014 - 2021 |
| Ocean tide levels | CATS2008 (Padman et al., 2008) | EPSG:3031 | 4000 | - |

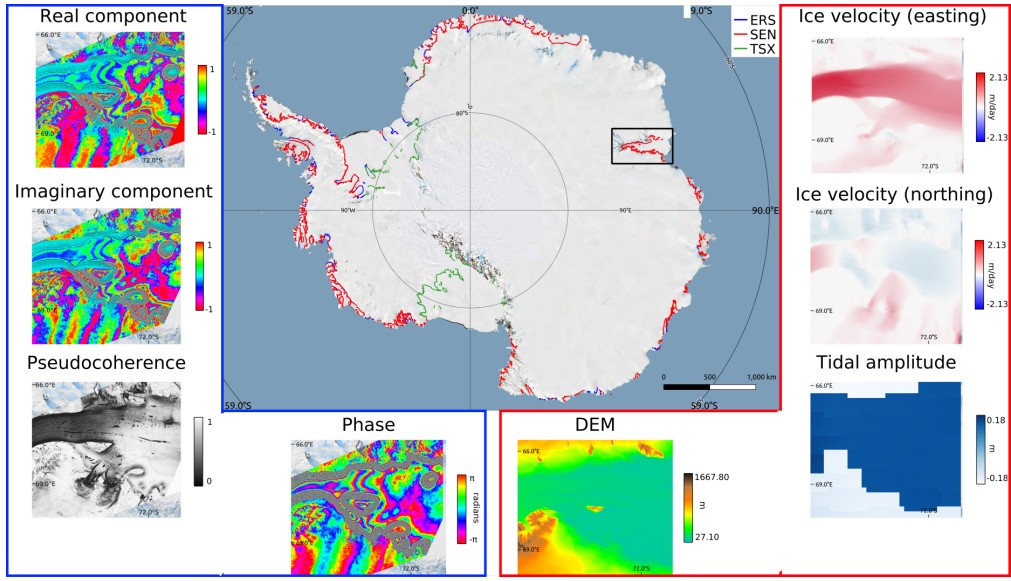

**Figure 2.** Center map: Manual grounding line delineations from in the AIS_cci GLL dataset (Groh, 2021). The legend shows the satellite missions used to derive the DInSAR interferograms. The rectangles surrounding the map show a sample training feature stack generated for the Amery Ice Shelf (black rectangle in the center map). The red polygon encloses non-interferometric features and the blue polygon encloses interferometric features.





## 4 Automatic delineation pipeline

### 4.1 Preprocessing

Each component of the feature stack needs to be prepared accordingly before entering the main delineation module. The AIS_cci GLL line geometries are converted into rasters with a pixel size of 100 m. The double-difference interferograms are resampled to correspond to their GL raster. The GL and interferogram rasters are then divided into tiles with 256 x 256 pixels dimensions and a 20% overlap in four directions. Subsequently, the features described in Section 3.2 are cropped and stacked to form a three-dimensional array. Any pixels with missing information are filled with the mean value of the features within
the crop. We normalize the non-interferometric features to the entire stack's global minimum and maximum values to ensure network stability during training. The feature tiles are partitioned into training, validation, and test sets (Section 5.1). Random flips, either along the horizontal or vertical axes, are applied to augment the training set, doubling the number of training samples. The individual preprocessing steps are illustrated in the corresponding block in Fig. 3.

### 4.2 GL delineation

We trained the Holistically-Nested Edge Detection (HED) (Xie and Tu, 2015) to delineate GLs on the interferograms. The network (Fig. 4) contains five convolution blocks in which every layer performs a padded convolution using a 3 x 3 kernel. The convolution blocks are separated by a max pool layer, which performs downsampling by a factor 2 on the tensors emerging from the convolution blocks. The final segmentation map is derived through a dynamically weighted combination of outputs from the last layer of each convolution block, which are initially upsampled to attain consistent dimensionality before concatenation.
To offset the class imbalance between the grounding line and background pixels, we used the weighted cross entropy loss function proposed by (Xie and Tu, 2015):

$$L(\hat{y}) = -\frac{|y_{gl}|}{|y|} \sum_{j \in y_{bg}} \log\left(1 - \hat{y_j}\right) - \frac{|y_{bg}|}{|y|} \sum_{j \in y_{gl}} \log \hat{y_j} \tag{1}$$

where $|y_{gl}|$ is the number of grounding line pixels and $|y_{bg}|$ is the number of background pixels and $|y|$ is the total number of pixels for one sample. The function weights the predicted probabilities of grounding line pixels by the fraction of background
pixels and vice versa for each tile.

We apply this loss function to each side output to enhance the learned features in the network's initial layers. This technique, called deep supervision, has been demonstrated to augment generalization and mitigate the challenge of vanishing gradients in segmentation tasks (Lee et al., 2015; Xie and Tu, 2015). Furthermore, the Rectified Linear Unit (ReLU) activation function is employed for every convolution layer, while the sigmoid function is applied to both the side outputs and the concatenated
output. Once the neural network has been trained with the specified parameters outlined in Section 5.1, we input the test samples into the network. This generates segmentation maps where each pixel denoted the probability of belonging to the grounding line class.





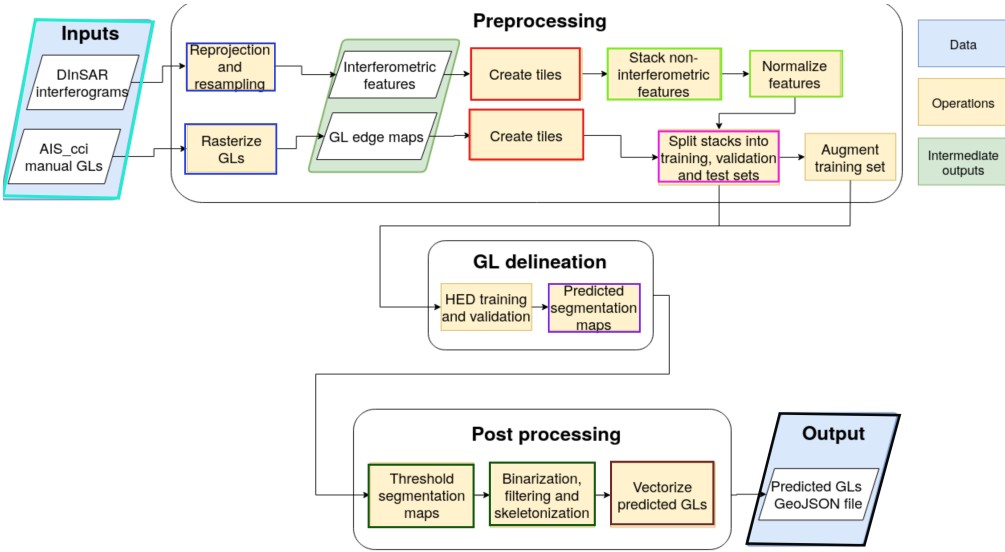

**Figure 3.** Flowchart outlining the operations in our automatic delineation pipeline. The colored borders of each step correspond to the illustrations depicted in Fig. 5.

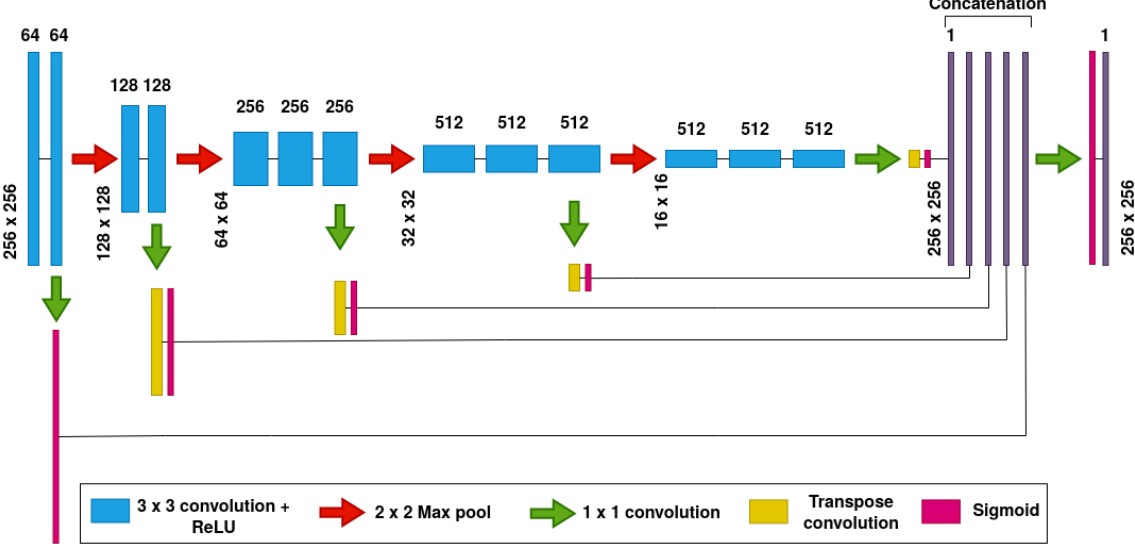

**Figure 4.** Architecture of the Holistically-Nested Edge Detection neural network (Xie and Tu, 2015).

## 4.3 Post-processing

The output segmentation maps from the delineation module are filtered to remove uncertain predictions. This filtering procedure comprises three stages. First, we convert the segmentation maps to binary values by applying a fixed threshold of 0.8. Pixels





with a value greater than or equal to this threshold are set to 1, while the rest are set to 0. We then apply a median filter with window size 3 to the binarized predictions to eliminate spurious branches. Finally, we apply a skeletonization algorithm (Zhang and Suen, 1984), which iteratively removes pixels until a one-pixel-wide skeleton remains. The skeletonized prediction rasters are pruned to remove spurious side branches using APIS from the PlantCV python library (Gehan et al., 2017) and then converted to line vectors, which are saved as a GeoJSON file. Fig. 3 shows the schematic for the automatic delineation pipeline. The output of the operations for an interferogram and its manual delineation is illustrated in Fig. 5.

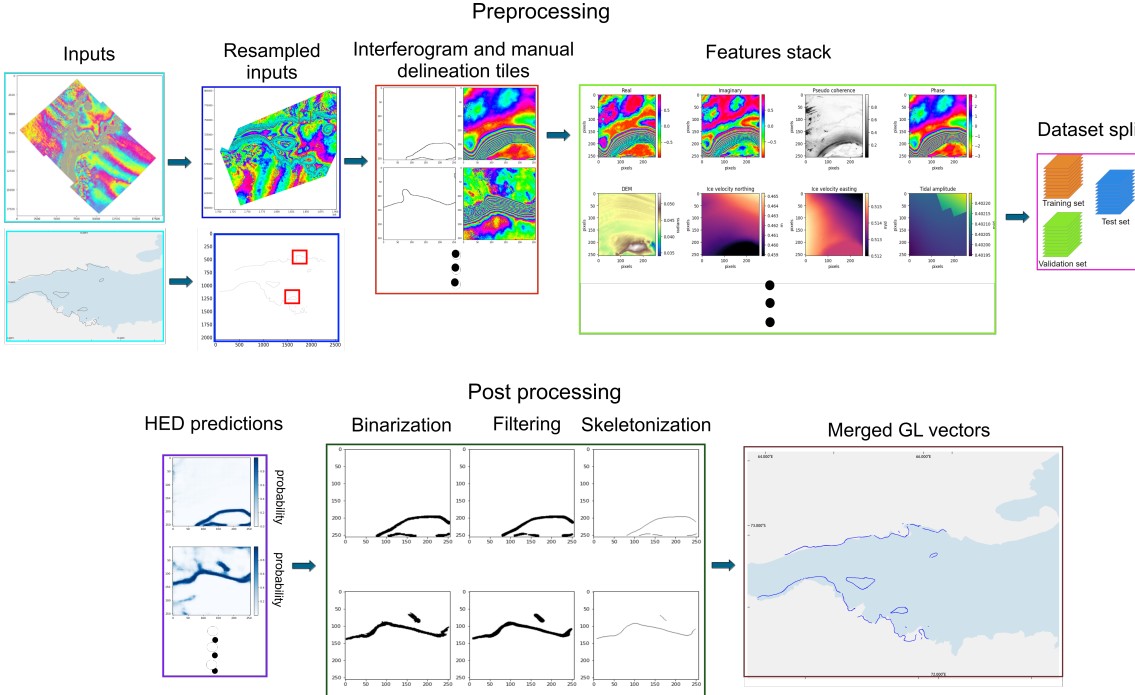

**Figure 5.** Illustration of the steps described in the delineation pipeline flowchart (Fig. 3) for an exemplary interferogram of the Amery Ice shelf (black rectangle in Fig. 2). The colours of the outlines of each image correspond to the respective highlighted operation in Fig. 3

## 5 Experiments and performance assessment

### 5.1 Training scheme

We trained our models on the NVIDIA A100 GPU with 80 GB high bandwidth memory for a maximum of 100 epochs and a batch size of 128 tiles. We used the Adam optimizer with parameters recommended by Kingma and Ba (2014) and a learning rate of $3 \times 10^{-4}$. We created two variants of our training, validation and test sets. In the in-sample dataset the manual GLs overlap spatially and not temporally. There are 4227 tiles in the training set, the validation set has 121 tiles, and the test set has 308 tiles. On the other hand, the out-of-sample dataset is split so there is no spatial overlap between the different sets.





In this case, the training set contains 4223 tiles, the validation set 118 tiles, and the test set 589 tiles. Both training sets were augmented by randomly flipping the tiles along the horizontal or vertical axes, doubling the number of samples. The neural network was implemented in the PyTorch Lightning framework (Falcon and The PyTorch Lightning team, 2019).

## 5.2 Feature importance

As mentioned in Section 3.2.1, we express the complex DInSAR interferograms as real and imaginary parts, i.e., the rectangular representation, as well as the pseudo coherence and phase components, i.e., the polar representation. We train two networks, one with rectangular interferometric features and the other with polar interferometric features in their training features stack, to determine the optimal representation of the interferogram for GL delineation.

Given the high dimensionality of our training dataset, we carried out several experiments to determine the importance of the non-interferometric features (enclosed in the red polygon in Fig. 2) to the delineation. Similar to Loebel et al. (2022), we report a relative feature importance by employing the leave-one-covariate-out (LOCO) inference method proposed by Lei et al. (2018). This technique involves training several models with different feature subsets. The importance of a feature is gauged by the performance of the model trained with this feature compared to the model in which this feature was ablated. Due to the correlation between the northing and easting ice velocity components, they are considered a single feature. All the network variants trained under this experiment contained the optimal subset of interferometric features, which was determined as an outcome of training the neural networks as described above. We used the samples of the in-sample dataset variant (Section 5.1, Fig. 6) and evaluated the performance on the respective test set.

## 5.3 Spatial transferability and generalization

To investigate HED's generalization capabilities, we used the network trained with rectangular features of the training samples from the in-sample dataset to delineate the test samples from the out-of-sample dataset (Fig. 7). We compared these delineations to those generated by HED, which was trained with the training samples of the out-of-sample dataset.

## 5.4 Performance evaluation metrics

We quantify the pixel-wise segmentation performance of HED with the Optimal Dataset $F_1$ score (ODS $F_1$) and Average Precision ($AP$). $F_1$ score (Eq. 2) is the harmonic mean of precision, $P$ and recall, $R$. The ODS $F_1$ score is then computed for multiple samples after converting the predictions to a binary map at a threshold value that yields the highest score. The $AP$ (Eq. 3) is the average of precision values for every threshold weighted by the corresponding increase in recall from the previous threshold.

$$F_1 = 2 \times \frac{P \times R}{P + R} \text{ with } P = \frac{T_p}{T_p + F_p} \text{ and } R = \frac{T_p}{T_p + F_n} \qquad (2)$$

where $T_p$ are true positive, $F_p$ are false positive and $F_n$ are false negative pixels.



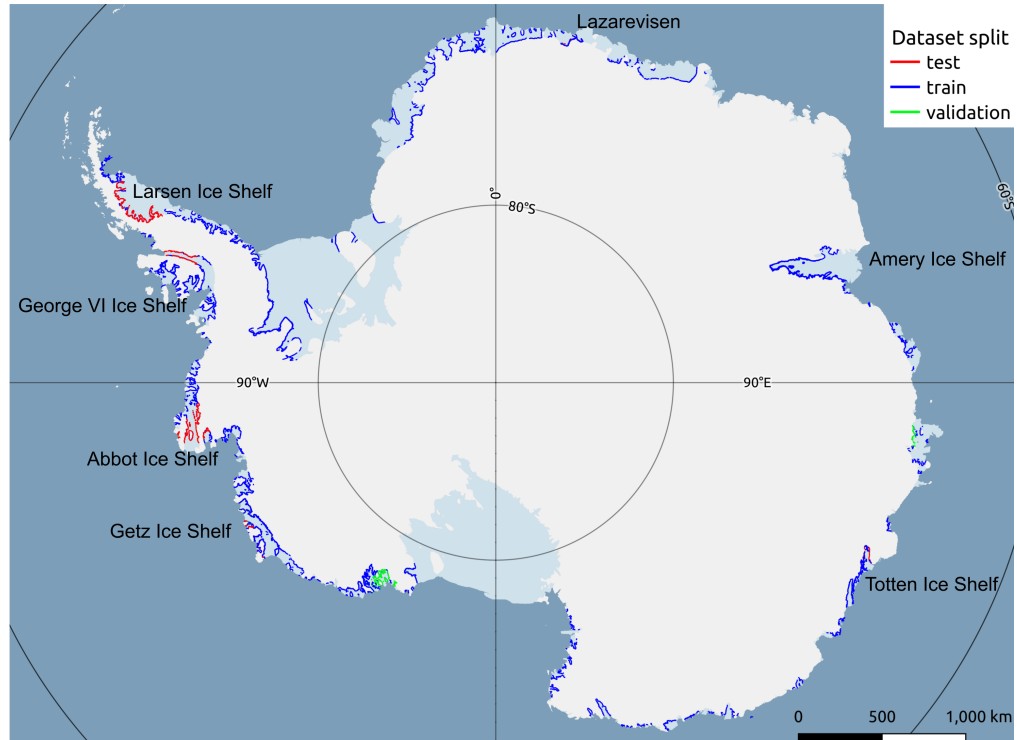

**Figure 6.** The spatial distribution of the AIS_cci GLLs (Fig. 2 center map) selected for the in-sample dataset, with the names of the glaciers and ice shelves of the test set displayed.

$$AP = \sum_n (R_n - R_{n-1}) \times P_n \qquad (3)$$

where $P_n$ and $R_n$ are the precision and recall at the $n$th threshold.

Our primary measure to evaluate the GL geometries obtained after post-processing the neural network segmentation maps is the metric for polygons and line segments (PoLiS) (Avbelj et al., 2014). For two polygons $A$ and $B$, the PoLiS distance is the average of distances between the vertices of one polygon to its closest point (which may not be a vertex) on the other polygon (Eq. 4). Since our network-generated and manual GL delineations are line geometries, we measure the Euclidean distance between points on the delineations from the neural network and the points of the corresponding manual delineations.

This metric is similar to the Mean Distance Error (MDE), which was first formally defined by Gourmelon et al. (2022), with the only difference being that the PoLiS distance is calculated for vector geometries while MDE is a pixel-based metric.

$$p(A, B) = \frac{1}{2|A|} \sum_{a \in A} \min_{b \in B} ||a - b||_2 + \frac{1}{2|B|} \sum_{b \in B} \min_{a \in A} ||b - a||_2 \qquad (4)$$



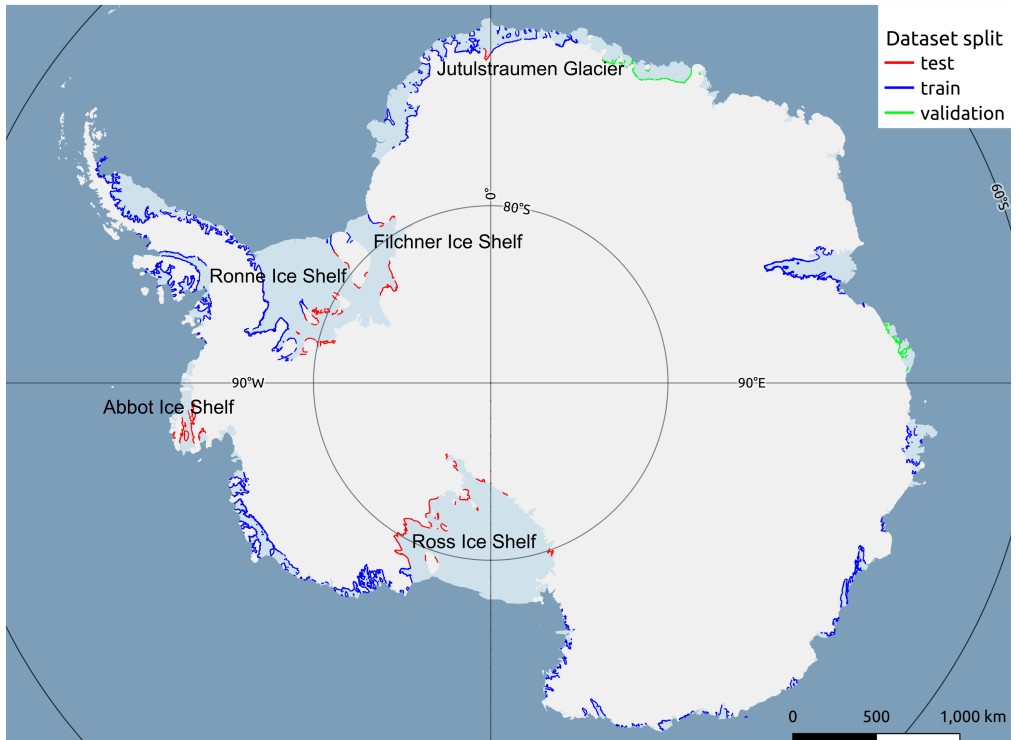

**Figure 7.** The spatial distribution of the out-of-sample dataset, with the names of the glaciers and ice shelves of the test set displayed.

where $|A|$ refers to the total number of grounding line points in the ground truth and $|B|$ is the total number of grounding line points in the network-generated GL.

We also computed the fraction of the length of manual GLs identified by the neural network as the coverage percentage (Eq. 5). Only the length of those network-generated GLs, which are at most the median PoLiS distance away from the manual GL, are considered, thereby excluding spurious detections from the calculation.

$$\text{Coverage} = \frac{\text{length of network-generated GL}}{\text{length of corresponding manual GL}} * 100\% \tag{5}$$

## 6 Results and discussion

For all conducted experiments, we report the metrics defined in Section 5.4 for the test set samples of the in-sample and out-of-sample datasets respectively. We present the mean and median PoLiS distances, the Median Absolute Deviation (MAD) as a measure of dispersion, and the average length of the network-generated GLs as a fraction of the length of the AIS_cci GLL lines. The ODS $F_1$ score and average precision $AP$ were calculated considering all the pixels of the test set.





## 6.1 Importance of interferometric features

Table 3 shows the metrics for the experiments described in Section 5.2. The networks 1 and 2 were trained with the rectangular and polar features and generated GLs with a median distance of less than 300 m from the manual GLs. A visual inspection of the network generated lines in Fig. 8 shows that both networks can follow the complex GL geometry to a large extent and seem to be able to distinguish between landward and seaward extents of the flexure zone. An example of this is seen in the GLs of the Abbot Ice Shelf test sample, where the networks correctly delineate the landward fringes despite the incorrect seaward manual delineation (green rectangle in the bottom row of Fig. 8). Unfortunately both networks still generate spurious branches and fail to detect parts of the grounding line which curve sharply (enclosed in the blue and black rectangles respectively in Fig 8), resulting in non-continuous GLs.

The average precision and ODS $F_1$ score for both experiments is very low, given the numerous false positives near the grounding line which is in agreement with the observations of Heidler et al. (2022a) and Gourmelon et al. (2022). Since the final GLs are only available after post-processing the segmentation maps from the network, the PoLiS distances are more representative of the delineation quality.

The mean distances between the network generated and the ground truth GLs are greater than their respective median distances due to sparse delineations on a few incoherent interferograms. Consequently, the coverage percentages are skewed, resulting in lower average coverage. A closer look at these challenging samples' high-resolution phase images (the insets in Figure 9) reveals areas which are close to decorrelation. The manual delineations for these interferograms are an example of the variability in the grounding line delineation performed by an operator. As previously observed by Mohajerani et al. (2021), the failure scenarios of the neural network are relatively consistent compared to those of a human expert.

## 6.2 Importance of non-interferometric features

Since the performance of the networks with the rectangular features is quantitatively and qualitatively better than the network trained with polar features, we proceeded to use the rectangular features in combination with various subsets of non-interferometric features.

The GLs delineated by HED, trained with the rectangular interferometric features and all the non-interferometric features, have a median PoLis distance of 186 m and cover nearly 80% of the manual GLs (network 3 in Table 3). The GLs delineated by this network are visualized for a few test samples in Fig. C1. Specifically, including the DEM in the training stack significantly boosts the network's performance, as evident from the low median distance of networks 4, 7, and 9 in Table 3. We attribute the improvement to the visibility of the break in slope feature $I_b$, which coarsely divides grounded ice from the ice shelf and can be beneficial for detecting the grounding line for fast-moving glaciers where it is difficult to obtain coherent interferograms. However, since the used DEM contains a mosaic of heights obtained by stacking several years of SAR acquisitions (Section 3.2.2), the break in slope may not always coincide with the GL captured in the DInSAR phase. In such cases, an over-reliance on the DEM can lead to incorrect delineations. An example is shown for the Cabinet Inlet test sample in Fig. 10 and Fig. C2, where the part of the GL enclosed in the blue rectangle incorrectly follows the steep slope of the DEM rather than the fringes



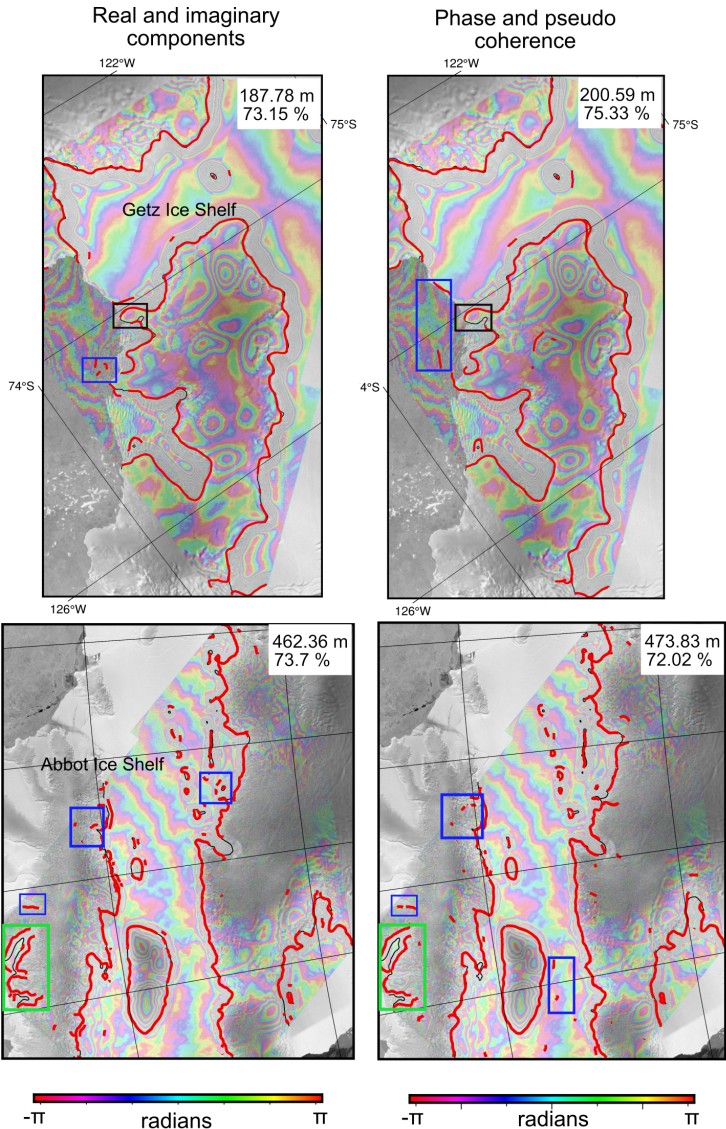

**Figure 8.** Visualization of the manual (black) and network-generated GLs (red) overlaid on three interferograms of the test dataset. The two lines are mostly overlapping. The columns correspond to networks 1 and 2 in Table 3. The legend in each subplot indicates the average PoliS distance between the manual and network GLs, and the fraction of length of the manual GL delineated by the network. The lines in the green rectangles are an example of an erroneous manual delineation, which was correctly delineated by the neural network. The blue rectangles show spurious detections and the black rectangles show parts of the GL that were not delineated.

of the flexure zone. In support of our findings, Heidler et al. (2022a) and Loebel et al. (2022) also observed that their models overfit to the DEM and Heidler et al. (2022a) cautions against their use for dynamic glaciers.





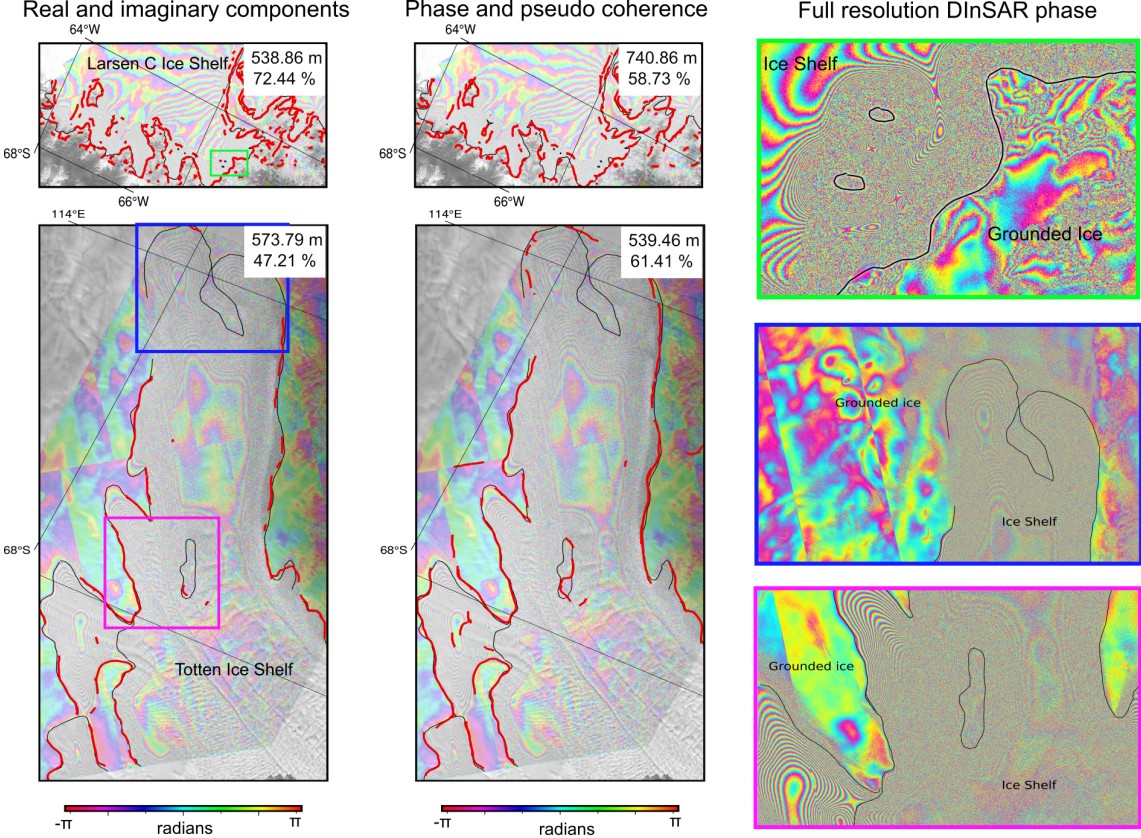

**Figure 9.** Visualization of the manual (black) and network-generated (red) GLs of test samples for which the networks 1 and 2 in Table 3 fail to adequately delineate. The full-resolution DInSAR phases correspond to the regions enclosed by the green, blue and pink rectangles.

When ice velocity and differential tide features are included together with the rectangular features in the training stack, the

network performance is the poorest (network 5 in Table 3, Fig. C4). However, when the ice velocity and differential tide are individually combined with the rectangular features, they contribute positively to the network performance, as indicated by the median distances, coverage percentages and the visual appearance of the delineations of networks 6 and 8 (Fig. C3). Adding the DEM to the feature stack further improves the delineations (networks 7 and 9, Fig. C4). When trained with these feature subsets, HED seems to rely more on the rectangular interferometric features or the DEM than the non-interferometric features.

**6.3 Inference on the out-of-sample dataset and undelineated interferograms**

The median distance of the delineations of the network trained on the out-of-sample dataset is slightly greater than the network trained on the in-sample dataset (Table 4). Visually, however, the delineations from both networks are comparable, as shown in Fig. 11. Even though the out-of-sample distribution did not contain any training sample that covered the Abbot Ice Shelf (Fig. 7), the delineation of the respective HED variant is very similar to the network trained on the in-sample dataset, in which all



**Table 3.** Numerical results for networks trained with different feature subsets as described in Section 5.2. The best performing network variant is highlighted in bold.

| Network | Features subset [m] | Median distance [m] | Mean distance [m] | MAD [%] | Mean coverage score | ODS F1 precision | Average |
|---|---|---|---|---|---|---|---|
| 1 | Real and imaginary components (rectangular features) | 276.0 | 433.08 | 148.15 | 69.58 | 0.17 | 0.080 |
| 2 | Phase & pseudo coherence (polar features) | 282.75 | 459.56 | 142.56 | 68.71 | 0.18 | 0.09 |
| 3 | **Real & imaginary components + non-interferometric features** | **186.0** | **289.19** | **84.25** | **78.6** | **0.21** | **0.11** |
| 4 | Real & imaginary components + DEM | 236.5 | 353.21 | 114.5 | 73.82 | 0.19 | 0.09 |
| 5 | Real & imaginary components + ice velocity + differential tide level | 862.0 | 1007.25 | 429.0 | 37.9 | 0.052 | 0.02 |
| 6 | Real & imaginary components + ice velocity | 314.0 | 446.42 | 163.43 | 68.87 | 0.15 | 0.08 |
| 7 | Real & imaginary components + DEM + differential tide level | 247.75 | 386.46 | 125.38 | 74.34 | 0.18 | 0.09 |
| 8 | Real & imaginary components + differential tide level | 253.25 | 384.8 | 120.63 | 72.07 | 0.17 | 0.09 |
| 9 | Real & imaginary components + DEM + ice velocity | 227.13 | 359.02 | 107.72 | 74.7 | 0.20 | 0.10 |

but one interferogram was a part of the training set. Still, the out-of-sample HED GLs are more fragmented and spurious than those of the in-sample network. The latter network perhaps benefited from seeing several interferograms for the same region in the training set (Marochov et al., 2021), and therefore, finds application in producing a time series of GLs for regions with a sufficient number of coherent interferograms.

**Table 4.** Numerical results for the experiments described in Section 5.3. The best performing network variant is highlighted in bold.

| Features subset | Median distance [m] | Mean distance [m] | MAD [m] | Mean coverage [%] | ODS F1 score | Average precision |
|---|---|---|---|---|---|---|
| **In-sample** | **269.5** | **459.48** | **141.86** | **65.94** | **0.17** | **0.08** |
| Out-of-sample | 316.0 | 505.66 | 176.25 | 65.18 | 0.13 | 0.05 |

We further tested the in-sample HED's spatial scalability by delineating interferograms covering several glaciers that drain
into the Ross Ice Shelf (Fig. 12a). These interferograms were not delineated by hand and were not a part of the AIS_cci
GLL product. Despite never having seen the interferograms during the training or validation stages, the network delineated



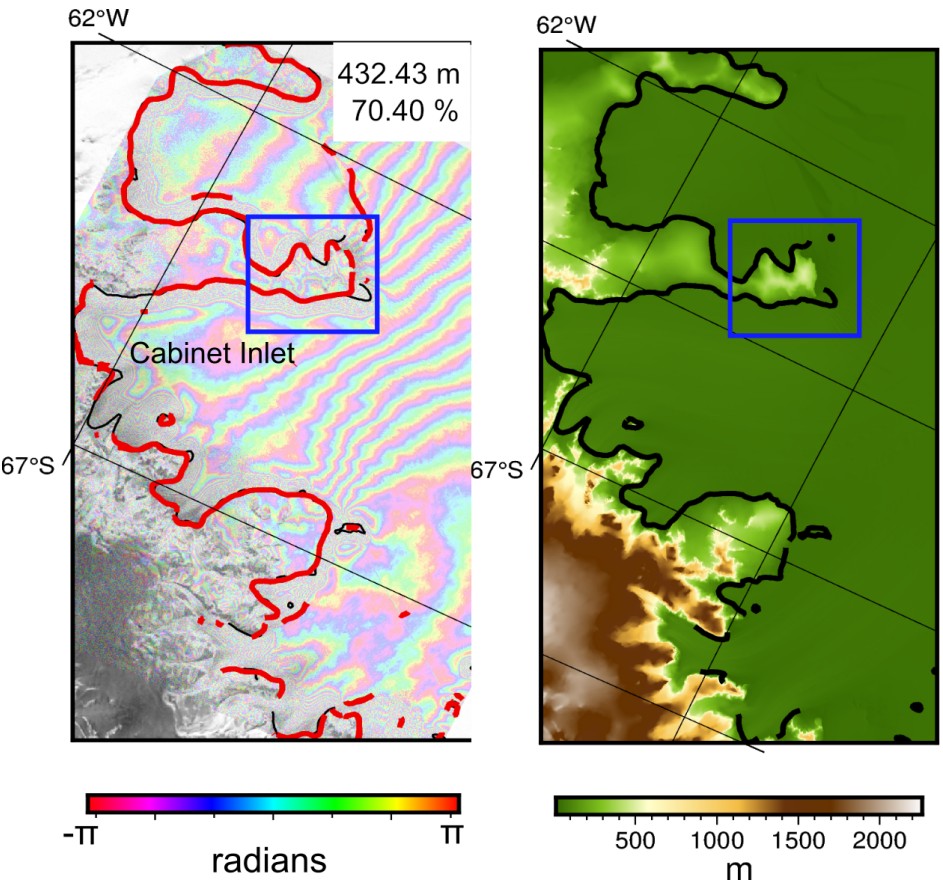

**Figure 10.** Visualization of a model variant overfitting to elevation data. The delineations of network 4 (red) in Table 3 for the Cabinet Inlet in Larsen C Ice Shelf. The right image shows the manually delineated GL (black) overlaid on the DEM feature for the sample. The delineation in the blue rectangle follows the change in slope in the DEM and not the landward-most fringe of the flexure zone.

the landward-most fringe and largely avoided delineating the decorrelated fringes of the Crary Ice Rise and Nimrod Glacier interferograms in Fig. 12c, d. The loose fringes in the interferograms of the Dickey Glacier and Nursery Glacier were not delineated. The network complemented existing manual delineations, reducing significant gaps and leading to a more complete

grounding line in this area.

## 7 Conclusions

We applied the HED deep neural network for automatically delineating grounding lines on DInSAR interferograms. The network was included as a module of our automatic GL delineation pipeline, which handles the preprocessing of DInSAR interferograms to make them suitable inputs for the network, training the network to generate GL segmentation maps and the

application of post-processing steps to obtain GL vector geometries. Our proposed network was trained on the AIS_cci GLL



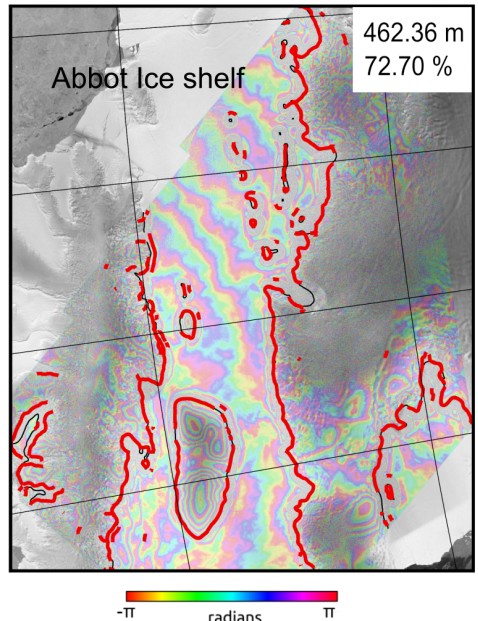
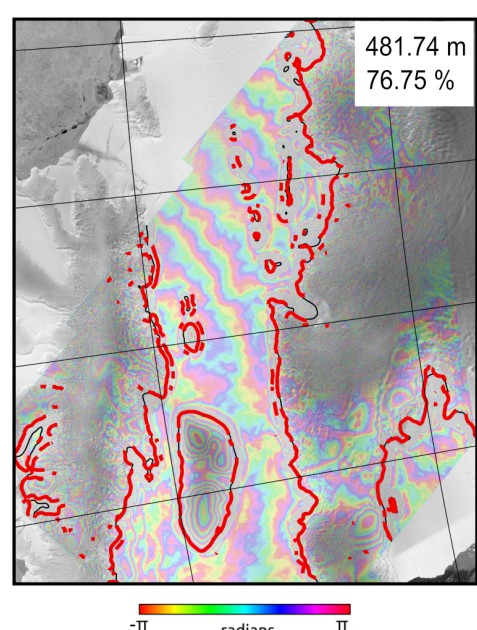

**Figure 11.** Delineations made by the HED (red) trained on the in-sample dataset and out-of-sample dataset for the Abbot Ice Shelf sample. Manual GLs are depicted in black.

dataset, which contains GL delineations that cover several outlet glaciers and ice streams bordered by ice shelves around the Antarctic Ice Sheet. Additional features, such as surface elevation information, northing and easting components of ice velocity and differential tide levels, were extracted from other datasets and were included in the HED training feature stack.

Our key finding from the feature ablation experiments is that the DEM helps refine the detection of the grounding line,
specifically for the cases where the DInSAR phase has poor coherence. However, the DEM may not reflect the dynamic nature of certain glaciers and could confound the network into making delineations that do not coincide with the landward side of the fringe belt captured in the DInSAR interferogram. Although they have a net positive contribution when combined with the DEM, training HED with the ice velocity and differential tide level features results in sparse and often spurious GL delineations. Therefore, we recommend using just the rectangular interferometric features to avoid spurious detections. The performance of
the HED variant trained with the rectangular features could be further improved by training an ensemble model, similar to the procedure employed by Herrmann et al. (2023), which we consider a potential future extension. We do not provide an estimate of the uncertainty of the network delineations, as this would require us to train neural networks based on Bayesian inference (Abdar et al., 2021), which is beyond the scope of this study.

We also demonstrated the ability of HED to delineate interferograms of previously unseen regions without retraining the
network, which enables the timely delineation of new interferograms without manual intervention. Our delineation pipeline is independent of the source of the SAR scenes and, therefore, can be used to delineate coherent DInSAR interferograms provided



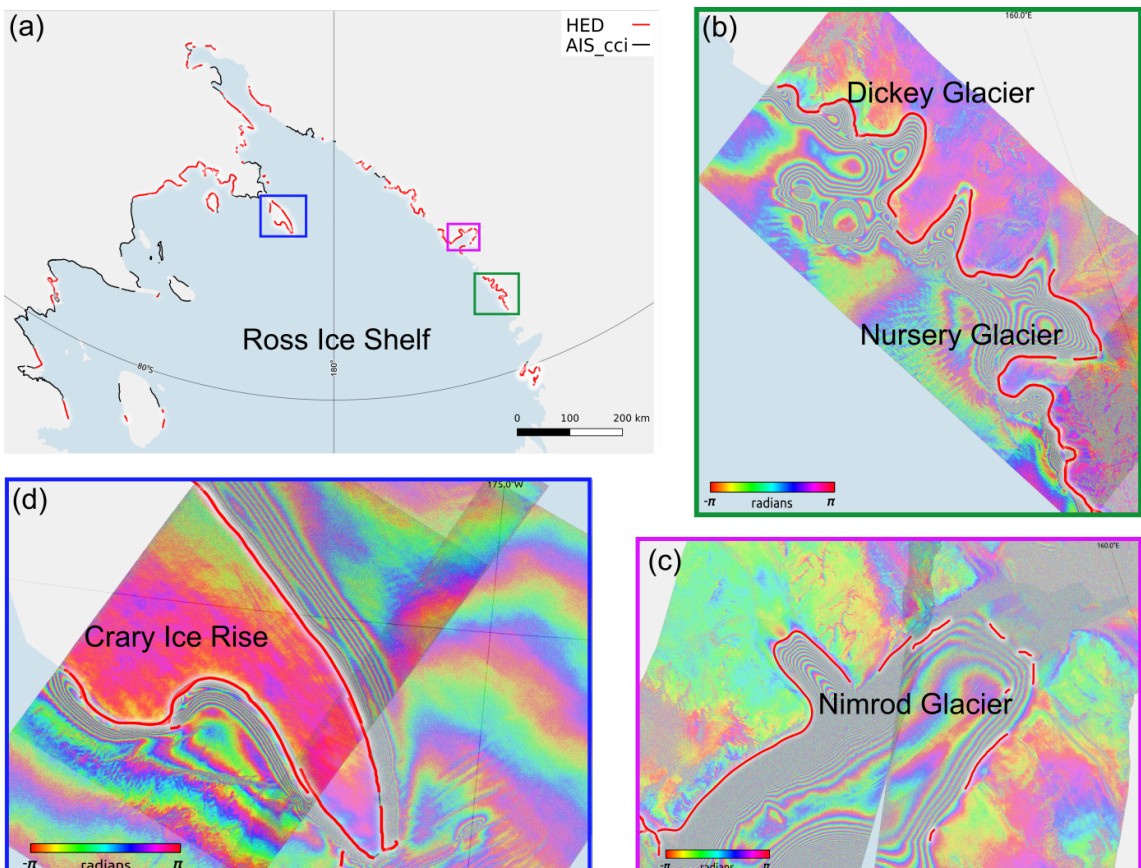

**Figure 12.** (a) AIS_cci GLL (black) and HED trained on the in-sample dataset (red) grounding lines for several glaciers situated around the Ross Ice Shelf. HED delineations for interferograms over (b) Dickey Glacier and Nursery Glacier, (c) Nimrod Glacier and (d) Crary Ice Rise. The discontinuous GLs for the Nimrod Glacier and Crary Ice Rise samples are from two spatially overlapping but temporally separate DInSARs. The interferograms, therefore, appear brighter in the overlapping regions.

by any spaceborne SAR mission. This is of particular interest for large interferometric datasets like e.g. those of the Sentinel-1 constellation and the soon to launch NASA-ISRO NISAR mission.



*Code and data availability.* The code, HED-generated GLs and a subset of the AIS_cci GLL product will be made available in a public
repository upon acceptance of this manuscript. Reviewers will be provided access to these utilities during the discussion phase.

*Author contributions.* SRT conceptualized the work, chose the neural network architecture, developed and implemented the automatic delineation pipeline, plotted the figures and wrote the manuscript. LK and DF supervised the work. KH provided his expertise on deep learning and training the neural network. DF, LK and KH made suggestions to the manuscript.

*Competing interests.* The contact author has declared that none of the authors has any competing interests.

*Acknowledgements.* We gratefully acknowledge the computational resources provided through the joint high-performance data analytics (HPDA) project "terrabyte" of the German Aerospace Center (DLR) and the Leibniz Supercomputing Center (LRZ), where the neural networks were fine tuned and final results were obtained. We also acknowledge the computational resources provided by the Helmholtz Association's Initiative and Networking Fund on the HAICORE@FZJ partition on which we first implemented and developed the automatic delineation pipeline. TerraSAR-X data were made available by DLR through the project HYD3056. This research was funded by
the Earth Observation Center, German Aerospace Center (DLR) Polar Monitor II and the ESA Antarctic Ice Sheet CCI (ESA/Contract No. 4000126813/18/I-NB) projects.



## Appendix A:  Automatic calving front delineation

Initial works tackled calving front delineation with classical image processing techniques such as edge enhancement, local thresholding (Sohn and Jezek, 1999), active contours (Klinger et al., 2011) and edge detection (Sohn and Jezek, 1999; Seale et al., 2011; Krieger and Floricioiu, 2017; Liu and Jezek, 2004) applied to optical and SAR images. However, these methods often required some degree of manual intervention, such as orienting images in the direction of glacier flow (Seale et al., 2011), specifying start and end points of the calving front (Krieger and Floricioiu, 2017), or providing an initial coastline position (Klinger et al., 2011), limiting their feasibility for large-scale implementation.

Mohajerani et al. (2019) developed the first deep-learning algorithm for CF delineation. They used the well-established UNet architecture (Ronneberger et al., 2015) to directly identify the CF pixels in Landsat imagery for glaciers in Greenland. Subsequent studies based on deep learning made improvements and modifications to UNet. They circumvented the class imbalance between calving front and background pixels by converting the problem into a multi-class segmentation task. They aimed to distinguish ice, non-ice and water regions in SAR images (Baumhoer et al., 2019; Zhang et al., 2019; Holzmann et al., 2021; Marochov et al., 2021; Periyasamy et al., 2022; Davari et al., 2022b) or multispectral images (Loebel et al., 2022). The CF was derived by tracing the boundary between the regions using post-processing techniques. These techniques do not directly apply to grounding line delineation, as current GL datasets do not indicate grounded and floating ice regions. Therefore, additional manual effort would be needed to create the appropriate dataset to train neural networks for such segmentation tasks.

Loebel et al. (2022) assessed the significance of topographical and textural features in addition to multispectral Landsat 8 images. They trained multiple models on subsets of input features and evaluated their contribution to CF detection. Holzmann et al. (2021) incorporated attention gates to the UNet architecture to guide the network to focus on the CF pixels and additionally used a loss function which applied a penalty proportional to the distance of the predicted pixels from the ground truth CF distances. Hartmann et al. (2021) added dropout layers to approximate a Deep Gaussian Process, thereby producing uncertainty maps of the network-generated CF delineation. Periyasamy et al. (2022) focused on optimizing the data preprocessing, data augmentation, loss function, architecture, normalization and dropout rate for the network. Davari et al. (2022b) used the Matthews correlation coefficient as an early stopping criterion. Marochov et al. (2021) took a different approach by classifying patches of training images that contained over 95% of pixels of the same class using a CNN based on VGG16 Simonyan and Zisserman (2014). These results were further utilized to train a Multilayer Perceptron (MLP) for pixel-wise classification into seven classes, distinguishing it from other studies in the field. These studies

Cheng et al. (2021), Heidler et al. (2022a) and Herrmann et al. (2023) combined edge detection and semantic segmentation to enhance classification at the ice-ocean boundary and derived precise calving fronts. Cheng et al. (2021) trained the DeepLabV3 network on SAR and multispectral images to generate both the calving front/background and the ice/ocean masks. Heidler et al. (2022a) merged two established CNNs, HED and UNet, designed for their respective tasks to delineate calving fronts in Sentinel-1 images. Herrmann et al. (2023) adapted the no new UNet framework (Isensee et al., 2021) and tested several learning regimes with modified network architectures, ranging from detecting just the calving front pixels to jointly segmenting the CF, glacier and non-glacier pixels.



Heidler et al. (2022b) moved away from semantic segmentation and instead iteratively modified an initial calving front delineation using a neural network variant of the active contours or Snakes algorithm (Kass et al., 1988). Their network outperformed the model developed by Cheng et al. (2021) while being trained on the same dataset. Davari et al. (2022a) reformulates the problem into a pixel-wise regression task, wherein they trained a UNet to predict the distance map transform of the CF pixels and applied several post-processing steps to retrieve the CF location. Herrmann et al. (2023) applied the no-new UNet architecture (Isensee et al., 2021) to the benchmark dataset compiled by Gourmelon et al. (2022). They explored the robustness of the network in several tasks, including the joint segmentation of the CF, glacier and non-glacier regions. Zhang et al. (2023) demonstrated the large-scale applicability of deep learning for calving front delineation by creating a comprehensive, fully automated processing pipeline that included automatic quality control and uncertainty quantification of network-derived lines.

Despite several similarities in grounding line and calving front delineation tasks, the discussed methods are not applicable out-of-the-box for GL delineation on DInSAR. Nonetheless, the existing approaches are a source of ideas for the further development of GL delineation with deep learning.

**Appendix B: Resampling wrapped phases**

In order to conserve the cyclic variation of phase, the amplitude and phase of the double difference interferograms were resampled separately. Due to missing SAR back scatter information for the interferograms used in the AIS_cci dataset, we used a unit amplitude component to obtain a complete complex polar representation of the interferograms. The polar components were transformed into real and imaginary parts (Fig. B1), which were resampled separately. These were transformed back to polar form to obtain resampled pseudo coherence and phase information (Fig. B2).



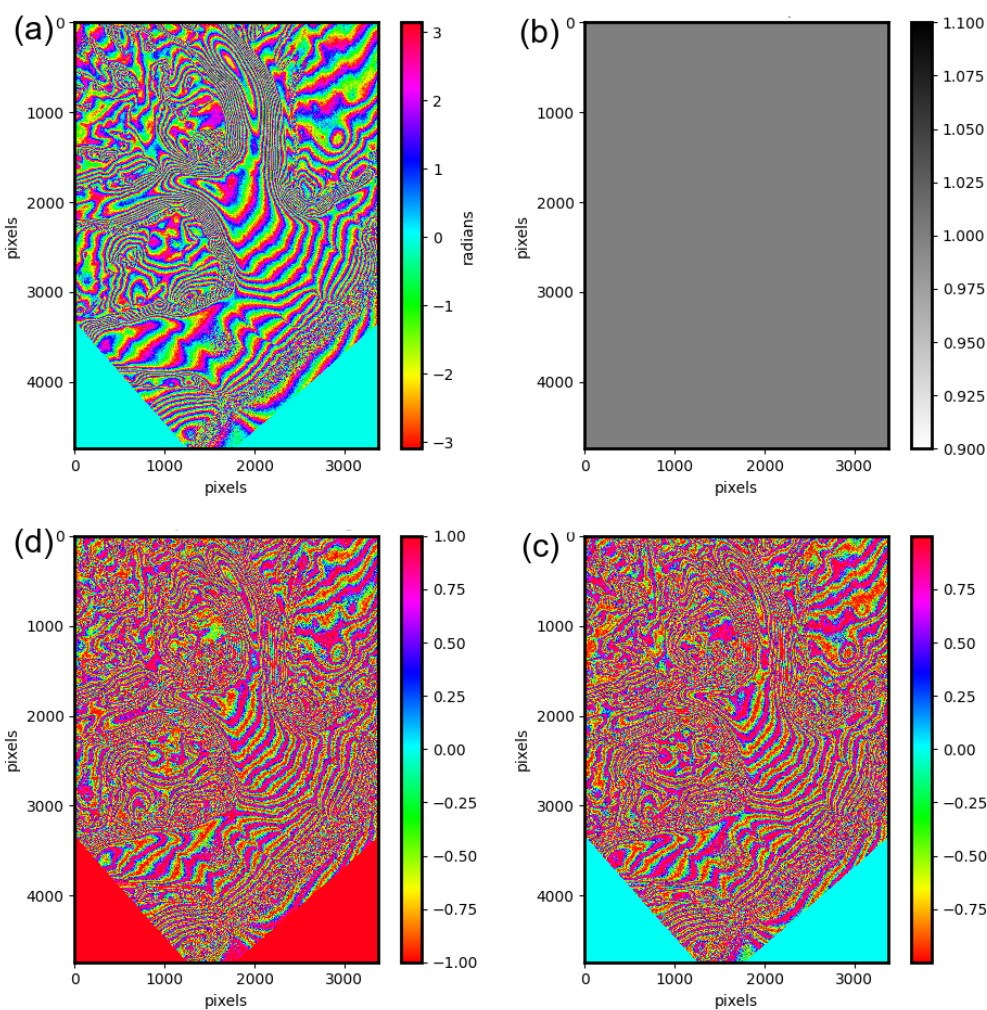

**Figure B1.** Phase preserving reprojection and resampling scheme illustrated for (a) sample ERS interferogram (b) the added unit amplitude component both in EPSG:4326 projection. Transformation from polar components to (c) imaginary and (d) real components.





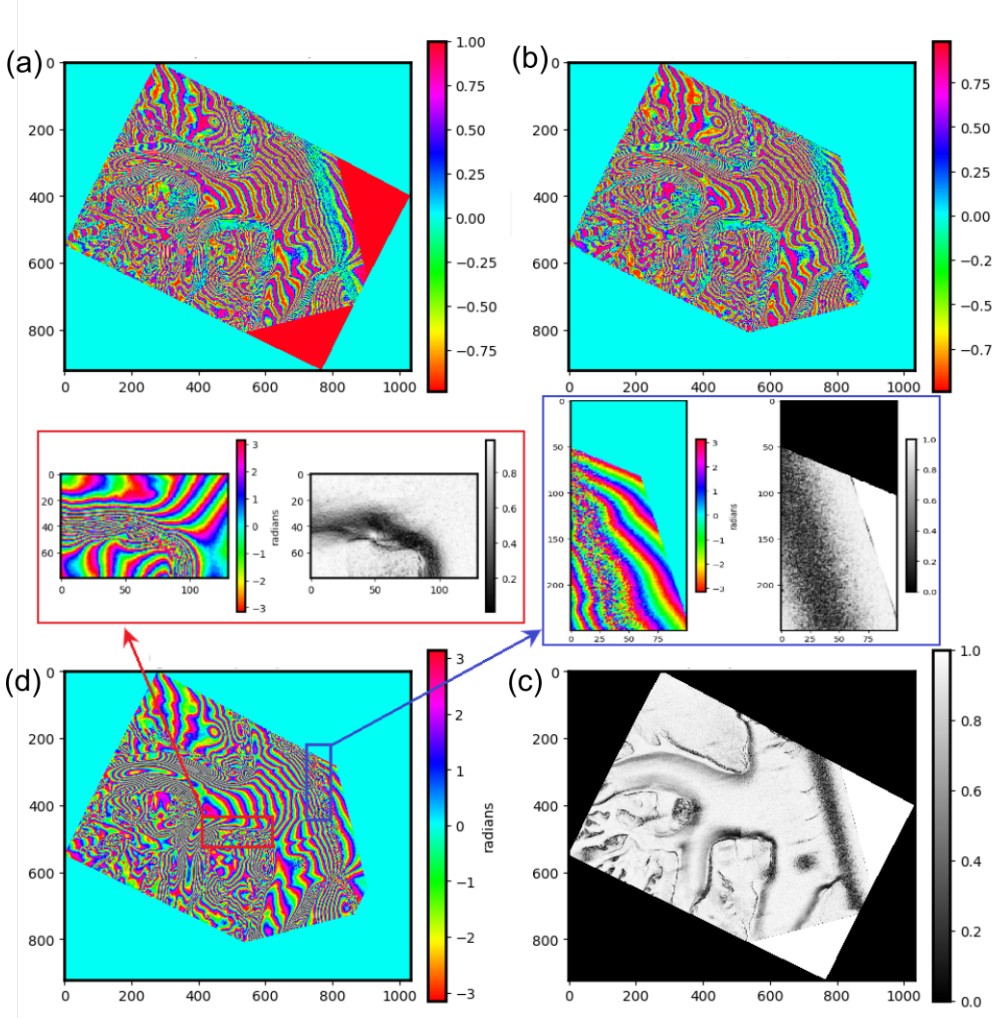

**Figure B2.** The resampled and reprojected (a) real and (b) imaginary interferometric components of the DInSAR phase sample shown in Fig. B1. Both were reprojected to EPSG:3031 projection with pixel size of 100m x 100m. These components are transformed back to (c) pesudo coherence and (d) phase. The insets in (d) show the variation of pseudo coherence for the cases where the phase is coherent but contains high frequency fringes (red rectangle) and where the phase is decorrelated (blue rectangle). In both cases, the pseudo coherence value is $< 0.4$

**Appendix C: Additional results**





**Figure C1.** Visualization of the GLs generated by networks 1, 2 and 3 (Table 3) for several test samples.





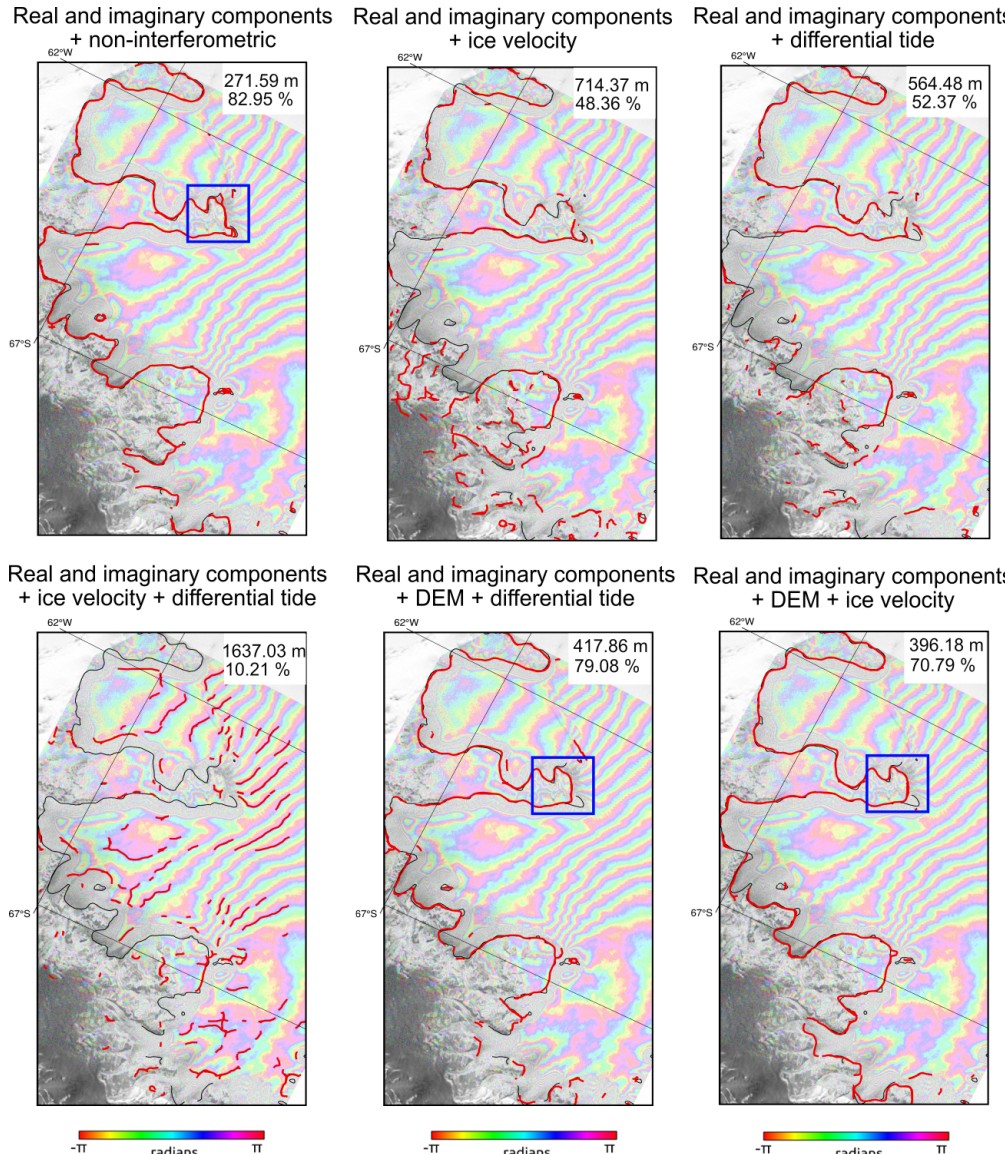

**Figure C2.** GL delineations of several network variants for Larsen C sample shown in Fig. 10. The part of the GLs enlcosed in the blue rectangles shows networks that contained the DEM in their training stack follows the change in slope in the DEM and not the landward-most fringe of the flexure zone.



**Figure C3.** Visualization of the GLs generated by networks 4, 6 and 8 (Table 3) for several test samples.





Figure C4. Visualization of the GLs generated by networks 5, 7 and 9 (Table 3) for several test samples.



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
