# Peer review of "Deep learning based automatic grounding line delineation in DInSAR interferograms"

_EGUsphere, 2024_

## Referee Comment (RC1)

**Review for "Deep learning based automatic grounding line delineation in DInSAR interferograms"**

**Summary:**

This manuscript used the Holistically-Nested Edge Detection (HED) neural network to delineate DInSAR interferograms automatically. Based on their experiments, they suggested using the rectangular interferometric features to avoid spurious detections. This manuscript demonstrated the ability of HED to delineate interferograms of previously unseen regions without retraining the network, which enables the timely delineation of new interferograms without manual intervention. And compared to the former paper published by Mohajerani et al., (2021), the authors used unentangled phases and pseudo-coherence, which could make a higher accuracy and more automated process. And they're supposed to be flexible enough to use the data from the three satellites, and then adjust the parameters for the resolution characteristics, the pseudo-coherence calculations, and the phase disentanglement, so that NISAR and other commercial satellites could be used as well. Overall, this manuscript is a well-written and innovative study in automatic groundling line delineation. There are several suggestions from my side that need to be addressed before the paper gets published.

**Major:**

1. The specific implementation details of error type analysis and uncertainty assessment are not mentioned in detail in this article. Error type analysis generally involves identifying and categorizing errors in model predictions because of the need to improve the model training process and the accuracy of the final output. Examples of unmentioned error classification include misidentifying non-baseline regions as baselines (false positives) or failing to identify true baseline regions (false negatives), and spatial errors, which are mislocalizations in the spatial distribution, such as shifts in baseline locations.

2. The uncertainty assessment is mainly concerned with calculating and presenting the confidence ranges of the model outputs, often using things like confidence intervals, or Bayesian statistics to estimate the uncertainty of the predictions. It seems that this manuscript doesn't explicitly use Bayesian networks or other statistical methods to express the uncertainty of the predictions, and it also doesn't go into detail about the types of errors that the model might produce, such as the quality of the input data, the structural limitations of the model and so on.

---

## Referee Comment (RC2)

Review for "*Deep learning based automatic grounding line delineation in DInSAR interferograms*".

This study introduces a novel deep learning framework utilizing Holistically-Nested Edge Detection (HED) to map Antarctic grounding lines (GLs), marking significant progress in the use of automatic algorithms for GL mapping from various remote sensing datasets. The network's demonstrated generalization capabilities highlight its potential for high-resolution temporal and spatial mapping of grounding lines, which is crucial for identifying grounding line migrations and understanding their dynamics. This timely study has the potential to make a valuable contribution to the field. However, several major concerns need to be addressed before I can recommend it for publication in The Cryosphere.

**Major Comments:**

1. Introduction and Related Work share many repetitive contents in terms of using non-deep-learning remote sensing methods in detecting GL. I recommend merging these two.
2. Dataset:
    a. The network-generated results have many spurious short line segments shown below (black lines – network GLs, red lines – AIS_cci GLs), any idea how to remove these inaccurate predictions when using the product?
    b. Uncertainty: In Mohajerani et al. (2021), they used the width of the vectorized contours as mapping uncertainty. With the threshold (0.8) scheme in your postprocessing, the mapping uncertainty can be easily achieved by applying different thresholds in extracting the grounding line.

[Figure]

3. I believe it is unnecessary to spend extensive effort discussing calving front mapping in this paper, as the primary focus is on detecting grounding lines. While grounding line detection shares similarities with glacier calving fronts, such as both being line segments, the input data sources are fundamentally different. Consequently, methods

effective for calving front detection may not be suitable for grounding line detection. It may be beneficial to mention that calving front edge detection inspired this research, but a detailed appendix reviewing various ML/DL methods for mapping calving fronts is unnecessary, especially since most referenced studies utilize UNET, unlike the edge detection approach in this research.

4. Additionally, you mention that Mohajerani et al. (2021) is the only study so far using a DL algorithm for mapping Antarctic grounding lines. However, there is no comparison between the models proposed in this study and those in Mohajerani et al. (2021). What are the benefits of using edge detection algorithms compared to the encoder-decoder architecture in Mohajerani et al. (2021)? How does your model's performance compare to that of Mohajerani et al. (2021)?

5. In-sample and out-of-sample variants:
    a. I am confused about creating two different variants of training/validation/test sets as in-sample and out-of-sample sets. I also wonder why these two variants are divided based on the spatial or temporal overlaps. The in-sample data are the datasets that model has access to during training and validation while out-of-sample data are used to test the model performance so it is a testing set, as such I don't understand why both in-sample and out-of-sample sets contain three individual training/validation/testing sets and why you need to train two different networks on these two datasets according to Section 6.3.
    b. In Table 4, I think the feature subset should be one of these interferometric/non-interferometric feature combinations listed in Table 3? Why here is In-sample or out-of-sample? When you train two networks for in-sample and out-of-sample datasets, which interferometric/non-interferometric feature combination did you use?
    c. From the paper itself, it seems you mainly used the in-sample training dataset to train the model and then evaluate the model performance on the in-sample and out-of-sample test sets, then what is the point of generating the out-of-sample training and validation sets?
    d. Table 3 shows the numerical results of different networks, however, here it only shows results for one test set, is it an in-sample or out-of-sample test set?

6. I am not convinced by Section 6.1. The importance of the interferometric features can only be proved by comparing them with networks trained with non-interferometric features. However, here you only compare networks 1 & 2, which are both trained with interferometric features.

7. Section 6.2 the importance of DEM (Line 270 and Figure 10):
    a. please include a detailed zoomed-in map of the interferogram inside the blue box. It seems the interferogram phase inside the blue box is decorrelated, so I won't be surprised that the network cannot map the correct GL. Also only giving one example with a small spatial extent is not representative.
    b. Have you checked the elevation change in Cabinet Inlet, is it a region undergoing significant elevation changes? If elevation is stable, I don't think you can attribute the wrong GLs to different DEM stacks.
    c. How to achieve the balance of including DEM to avoid over-reliance?

8. Section 6.3;
    a. As mentioned above, I don't understand why compile two different in-sample/out-of-sample sets and train two networks. If you combine the insample and out-of-sample sets into one dataset, won't this greatly increase the training samples and improve the model performance?

  b. You evaluate the in-sample trained model performance on the unseen Ross Ice Shelf interferogram by using Figure 12, however the discussion on the prediction quality is limited. Most GZ regions in Ross Ice Shelf are stable, I would like to see a distance deviation map between the AIS_cci GL and the network-generated GLs in Ross Ice Shelf to demonstrate the performance. If there are large deviations, please consider explaining 1) what are causing the large deviation? 2) which dataset is correct? 3) how can you further improve these results?

  c. In addition, I am curious to know what new GL information you can provide by using your approach. What is the implication of using your model in mapping the GLs and improving our understanding of the GL migrations?

9. Figures:
  a. Please consider labeling all the subplots in each figure, and adding a subplot to show the ROI location in Antarctica.

  b. Figure 8, it is impossible to visually compare the differences between GL predictions from these two networks given the current presentation format. I suggest plotting the spatial deviations between the network predictions so we can directly see where and how much these two are deviating from each other. Again, there are multiple ways to visualize this difference.

  c. Same problem with Figure 9:
    • Cross-referencing the three inset figures by just coloring the subplot figure frames is not helpful.
    • On Larsen C Ice Shelf, it is impossible to see the details of network-generated GLs inside the green box in the first subplot.
    • The plotting extent cut out the GLs in Totten main glacier stream, you need to expand the spatial extent.
    • Why not also plot the three inset boxes in the second column?
    • In the final column, you present the zoomed-in interferograms and show the manual GLs, why not plot the network-generated GLs from these two different networks so we will know the different performances of these two networks in Totten?

  d. Figure 11:
    • It's difficult to compare these two outputs without putting them in the same figure or providing a distance deviation map.
    • You have done an Antarctica-scale evaluation, why not include a comparison map for the whole ice sheet?

**Technical Comments:**

Line 15: provide the mass change uncertainty for both ice sheets.

Line 25-50: these three paragraphs need restructuring:

• Grounding line itself is a subglacial feature, please elaborate why detecting these two features is challenging and why different (surface) features can be used as proxies for the grounding line.

- You first cite Brunt et al., 2011 to say that existing methods detect grounding line proxies, then talk about using ice-penetrating radar in detecting true grounding line G which is a subglacial feature. The logic here is problematic.

Line 51: it is 'grounding line' not 'grounded line'.

Line 54: where is this research 'Ramanath Tarekere, 2022' published?

Line 63-64: ICESat laser altimetry has also been used in generating grounding zone products manually by Fricker et al. (2006, 2009) and Brunt et al. (2010, 2011).

Line 65: I see what you are trying to say here – emphasizing DL method does not need manual intervention compared to other methods. However, I find it a bit confusing to follow the logic. Having read the first sentence, I would expect to know the research progress in using DL methods in detecting GZ, but here you directly dive into model inversion and ICESat-2 methods.

Line 74-79: In addition to laser altimetry, there are several studies that have used CryoSat-2 radar altimetry in mapping GZ automatically, such as Dawson and Bamber (2017, 2020), and Hogg et al. (2018).

Line 138: the pyTMD should be cited as Sutterley et al. (2017). Check https://pytmd.readthedocs.io/en/latest/getting_started/Citations.html

Line 175: how did you determine 0.8 as the threshold?

Line 278-279: can you explain more about this claim? Given the current evidence in this section, I don't follow how you can claim that HED relies more on the rectangular interferometric features or DEM than the non-interferometric features.

Figure 2: It should be differential tidal amplitude.

Figure 5: I am confused about this figure:

- The subplot in the second row of the second column 'Resample Inputs', what are these two red boxes? Are these two different sampling locations that correspond to two different interferogram subsets in the third column? Also, what is the meaning of those three dots?
- I suggest replotting this figure to make it as clear as possible.

---

## Author Comment (AC1)

**Deep learning based automatic grounding line delineation in DInSAR interferograms**

Sindhu Ramanath Tarekere[1], Lukas Krieger[1], Dana Floricioiu[1], and Konrad Heidler[2]

[1]Remote Sensing Technology Institute, German Aerospace Center, Oberpfaffenhofen Germany
[2]Data Science in Earth Observation, Technical University of Munich, Germany

**Correspondence:** Sindhu Ramanath Tarekere (sindhu.ramanathtarekere@dlr.de)

**Authors' response to review comments (https://doi.org/10.5194/egusphere-2024-223-RC1)**

We thank the reviewer for providing their expertise and time in reviewing our work. We plan to revise our manuscript under consideration of their comments and suggestions. Below are point-by-point replies to the comments: reviewer comments are in black and the authors' response in blue.

1. The specific implementation details of error type analysis and uncertainty assessment are not mentioned in detail in this article. Error type analysis generally involves identifying and categorizing errors in model predictions because of the need to improve the model training process and the accuracy of the final output. Examples of unmentioned error classification include misidentifying non-baseline regions as baselines (false positives) or failing to identify true baseline regions (false negatives), and spatial errors, which are mislocalizations in the spatial distribution, such as shifts in baseline locations.

   We will provide the false positive and false negative detection rates on the test set in addition to the results provided in Section 6 of the manuscript. The median and mean deviations of the network delineated grounding lines from the manual delineations (Table 3 and 4 in the manuscript) is equivalent to the above mentioned mislocalization error.

2. The uncertainty assessment is mainly concerned with calculating and presenting the confidence ranges of the model outputs, often using things like confidence intervals, or Bayesian statistics to estimate the uncertainty of the predictions. It seems that this manuscript doesn't explicitly use Bayesian networks or other statistical methods to express the uncertainty of the predictions, and it also doesn't go into detail about the types of errors that the model might produce, such as the quality of the input data, the structural limitations of the model and so on.

   We will quantify the uncertainty of networks 1 and 3 (refer to Table 3 in the manuscript) by training an ensemble (Lakshminarayanan et al., 2017), (Valdenegro-Toro, 2019) of five networks each. The ensemble is built by initialising each network with a different set of random weights and by re-shuffling the training samples. We will provide the ensemble grounding line which is derived as the mean of the grounding lines of the individual members and 95% confidence intervals for test set samples. We will also include visualisations of the confidence interval as a buffer around the network delineations. The sub-categorisation of predictive errors into model and data uncertainties is a vast and active

field of research. Currently, no consistent procedures or protocols are defined to estimate these errors (Gawlikowski et al., 2023). Therefore, an in-depth error quantification is beyond the scope of our work. We will add the uncertainties for the DEM, ice velocity and tide levels provided by the original sources to Table 2.

**References**

Gawlikowski, J., Tassi, C. R. N., Ali, M., Lee, J., Humt, M., Feng, J., Kruspe, A., Triebel, R., Jung, P., Roscher, R., et al.: A survey of uncertainty in deep neural networks, Artificial Intelligence Review, 56, 1513–1589, https://doi.org/10.1007/s10462-023-10562-9, 2023.

Lakshminarayanan, B., Pritzel, A., and Blundell, C.: Simple and Scalable Predictive Uncertainty Estimation using Deep Ensembles, in: Advances in Neural Information Processing Systems, edited by Guyon, I., Luxburg, U. V., Bengio, S., Wallach, H., Fergus, R., Vishwanathan, S., and Garnett, R., vol. 30, Curran Associates, Inc., https://proceedings.neurips.cc/paper_files/paper/2017/file/9ef2ed4b7fd2c810847ffa5fa85bce38-Paper.pdf, 2017.

Valdenegro-Toro, M.: Deep sub-ensembles for fast uncertainty estimation in image classification, arXiv preprint, https://doi.org/10.48550/arXiv.1910.08168, 2019.

---

## Author Response (AR1)

**Deep learning based automatic grounding line delineation in DInSAR interferograms**

Sindhu Ramanath[1,2], Lukas Krieger[1], Dana Floricioiu[1], Codruț-Andrei Diaconu[1,2], and Konrad Heidler[2]

[1]Remote Sensing Technology Institute, German Aerospace Center, Oberpfaffenhofen Germany
[2]School of Engineering and Design, Technical University of Munich, Germany

**Correspondence:** Sindhu Ramanath (sindhu.ramanathtarekere@dlr.de)

**Authors' response to review comments (https://doi.org/10.5194/egusphere-2024-223-RC1) and https://doi.org/10.5194/egusphere-2024-223-RC2)**

We thank the reviewer for providing their expertise and time to review our work. We revised our manuscript after considering their comments and suggestions. The major changes that have been incorporated into the manuscript are summarized below:

1. 'Tarekere' was dropped from the last name of the primary author.

2. Codrut-Andrei Diaconu has been added as a co-author. He contributed to the uncertainty quantification of the neural network.

3. University affiliation has been added to Sindhu Ramanath and Codrut-Andrei Diaconu. Konrad Heidler's affiliation was corrected to [2].

4. Section 2 Related Works has been removed. The content has been merged into Section 1.

5. We added the uncertainty of all the datasets listed in Table 2.

6. Section 5 Estimation of predictive uncertainty was added. We now report the metrics of the ensemble in the abstract and conclusion.

7. Figures:

   (a) Minor changes were made to Fig. 3 and 5 as suggested by reviewers.

   (b) Figure 6 now clearly illustrates the difference between temporal and spatial dataset variants.

   (c) Figures 7, 8, 9, 10, B1, B2 and B3 contain an inset showing the ROI in Antarctica as suggested by reviewers.

   (d) The colors of the HED and AIS_cci GLs have been inverted in Fig. 10.

   (e) Figure 12 has been added to illustrate the concept of uncertainty buffers.

   (f) Figure 13 provides an overview of the region-wise performance of the ensemble.

Listed below are point-by-point replies to the comments: reviewer comments are in black, the authors' response in blue and snippets from the original preprint in red. Citations from the revised manuscript and additional replies to the comments are in light blue.

**Response to RC1 (https://doi.org/10.5194/egusphere-2024-223-RC1)**

1. The specific implementation details of error type analysis and uncertainty assessment are not mentioned in detail in this article. Error type analysis generally involves identifying and categorizing errors in model predictions because of the need to improve the model training process and the accuracy of the final output. Examples of unmentioned error classification include misidentifying non-baseline regions as baselines (false positives) or failing to identify true baseline regions (false negatives), and spatial errors, which are mislocalizations in the spatial distribution, such as shifts in baseline locations.

   We will provide the false positive and false negative detection rates on the test set in addition to the results provided in Section 6 of the manuscript. The median and mean deviations of the network delineated grounding lines from the manual delineations (Table 3 and 4 in the manuscript) is equivalent to the above mentioned mislocalization error.

   We have not included the false positive and false negative detection rates as these are encompassed in the average precision and are not particularly informative about the network performance.

2. The uncertainty assessment is mainly concerned with calculating and presenting the confidence ranges of the model outputs, often using things like confidence intervals, or Bayesian statistics to estimate the uncertainty of the predictions. It seems that this manuscript doesn't explicitly use Bayesian networks or other statistical methods to express the uncertainty of the predictions, and it also doesn't go into detail about the types of errors that the model might produce, such as the quality of the input data, the structural limitations of the model and so on.

   We will quantify the uncertainty of networks 1 and 3 (refer to Table 3 in the manuscript) by training an ensemble (Lakshminarayanan et al., 2017), (Valdenegro-Toro, 2019) of five networks each. The ensemble is built by initialising each network with a different set of random weights and by re-shuffling the training samples. We will provide the ensemble grounding line which is derived as the mean of the grounding lines of the individual members and 95% confidence intervals for test set samples. We will also include visualisations of the confidence interval as a buffer around the network delineations. The sub-categorisation of predictive errors into model and data uncertainties is a vast and active field of research. Currently, no consistent procedures or protocols are defined to estimate these errors (Gawlikowski et al., 2023). Therefore, an in-depth error quantification is beyond the scope of our work. We will add the uncertainties for the DEM, ice velocity and tide levels provided by the original sources to Table 2.

   Section 5 Estimation of predictive uncertainty discusses the procedure and shows the results of the model uncertainty for networks 1 and 3. We have added the uncertainty of the datasets which we used as training features to Table 2.

**Response to RC2 https://doi.org/10.5194/egusphere-2024-223-RC2)**

**Major comments**

1. Introduction and Related Work share many repetitive contents regarding non deep-learning remote sensing methods in detecting GL. I recommend merging these two.

   We agree that these sections can be merged. We will condense the content of Related Work and include it after line 48 in the manuscript.

   As suggested, we have merged the related work into the introduction.

2. Dataset:

   (a) The network-generated results have many spurious short line segments shown below (black lines – network GLs, red lines – AIS_cci GLs), any idea how to remove these inaccurate predictions when using the product?

   We have a python function $remove\_outliers$ (https://gitlab.dlr.de/rama_si/automatic_gll_delineation/-/blob/main/postprocess/vectorize.py?ref_type=heads) which filters out spurious lines which lie outside of a user given buffer (buffer distance in metres) around a reference GL (also user given). Please note that the accuracy of the routine is dependent on the chosen buffer distance and completeness of the lines in the reference GL dataset.

   Currently, this function is run as a part of our postprocessing procedure, resulting in unfiltered and filtered network delineations. The unfiltered lines were used for all the visualizations and metric calculations in the manuscript. We will provide both the unfiltered and filtered lines of the ensemble in our Zenodo repository (https://doi.org/10.5281/zenodo.11277696), along with a short description of the filtering routine.

   (b) Uncertainty: In Mohajerani et al. (2021), they used the width of the vectorized contours as mapping uncertainty. With the threshold (0.8) scheme in your postprocessing, the mapping uncertainty can be easily achieved by applying different thresholds in extracting the grounding line.

   The last layer of the neural network proposed by Mohajerani et al. (2021) uses the Sigmoid function to produce scores in the range $[0,1] \in \mathbf{R}$ for each pixel. To our understanding, the contours are constructed by choosing those pixels that the network scores above 0.3. We believe this to be a proxy for the actual neural network uncertainty, as these scores are not associated with the model uncertainty (Gawlikowski et al., 2023). Similarly, the threshold (0.8) in our postprocessing scheme is merely a criterion for choosing the pixels the neural network scores 0.8 and above. Therefore, varying this threshold would not quantify the uncertainty of our proposed neural network. Instead, we will train an ensemble of five neural networks and provide the mean network delineation as well as the 95% confidence interval for each sample. Please refer to our response to reviewer 1 for more details on uncertainty quantification https://doi.org/10.5194/egusphere-2024-223-AC1.

   Section 5 Estimation of predictive uncertainty discusses the procedure for uncertainty estimation. We computed $\pm$ 1 standard deviation buffers for each sample.

3. I believe it is unnecessary to spend extensive effort discussing calving front mapping in this paper, as the primary focus is on detecting grounding lines. While grounding line detection shares similarities with glacier calving fronts, such as both being line segments, the input data sources are fundamentally different. Consequently, methods effective for calving front detection may not be suitable for grounding line detection. It may be beneficial to mention that calving front edge detection inspired this research, but a detailed appendix reviewing various ML/DL methods for mapping calving fronts is unnecessary, especially since most referenced studies utilize UNET, unlike the edge detection approach in this research.

   As you suggest, we will cite a few important references for deep learning-based calving front delineation and remove the detailed survey from the appendix.

   Section 2 Related works has been merged with Section 1 Introduction. The survey about calving front delineations has been removed from the appendix.

4. Additionally, you mention that Mohajerani et al. (2021) is the only study so far using a DL algorithm for mapping Antarctic grounding lines. However, there is no comparison between the models proposed in this study and those in Mohajerani et al. (2021) . What are the benefits of using edge detection algorithms compared to the encoder-decoder architecture in Mohajerani et al. (2021)? How does your model's performance compare to that of Mohajerani et al. (2021)?

   We have compared the performance of the network proposed by Mohajerani et al. (2021) to ours but chose not to add it to the first draft of the manuscript due to some limitations. First, The training dataset for the study was not published. Second, we could not reproduce their computing environment. We implemented their network architecture and trained it on our dataset, but did not achieve the performance stated in Mohajerani et al. (2021) and therefore do not think this is a fair comparison. Nevertheless, we will add the experimental results in our revised manuscript while mentioning the above-stated drawbacks. We also performed a few experiments in which we trained a UNet on the same features stack (Ramanath Tarekere, 2022) and found our targeted edge detection neural network to work better for this task. We will mention these experiments in the discussion.

   We trained several versions of the DeeplabV3+ neural network on our dataset but unfortunately still could not generate grounding lines comparable to those shown in Mohajerani et al. (2021). The performance of neural networks is highly dependent on the quality of the training data and requires a non-trivial amount of time to tune its hyperparameters to achieve state-of-the art performance. Since we do not claim that our network performs better than DeeplabV3+, we believe that investing significant additional time into refining this model would not provide meaningful benefits to the experiments described in the manuscript. While we agree that such a comparison would be interesting, it does not really fit into the narrative, which is to investigate the influence of non-interferometric features on grounding line delineation.

5. In-sample and out-of-sample variants:

   (a) I am confused about creating two different variants of training/validation/test sets as in-sample and out-of-sample sets. I also wonder why these two variants are divided based on the spatial or temporal overlaps. The in-sample data

are the datasets that the model has access to during training and validation, while out-of-sample data are used to test the model performance, so it is a testing set, as such I don't understand why both in-sample and out-of-sample sets contain three individual training/validation/testing sets and why you need to train two different networks on these two datasets according to Section 6.3.

Thank you for this comment. We agree that we must adequately clarify the distinction between in-sample and out-of-sample splits. We considered two different distributions of the AIS_cci dataset, i.e., they are simply two different ways of splitting the same dataset into training, validation and test sets. The samples for any ROI in the out-of-sample split belong exclusively to either the training, validation or testing sets. Conversely, the in-sample split contains training and test samples or training and validation samples with different epochs from the same ROI. However, there are samples for certain ROIs that belong exclusively to the training set. Figure 1 of this response shows examples of several ROIs.

The purpose of training the networks on these two splits was to determine how well they could delineate interferograms outside their training sets' spatial and temporal domains. Such evaluations have been performed for deep learning models that were developed for other cryosphere-related applications which deal with high spatiotemporal varying data, such as glacier mass balance (Guidicelli et al., 2023) and calving front delineation (Gourmelon et al., 2022), (Herrmann et al., 2023). Our results in Section 6.3 indicate a benefit in training the network with an in-sample dataset:

Line 284: "Even though the out-of-sample distribution did not contain any training sample that covered the Abbot Ice Shelf (Fig.7), the delineation of the respective HED variant is very similar to the network trained on the in-sample dataset, in which all but one interferogram was a part of the training set. Still, the out-of-sample HED GLs are more fragmented and spurious than those of the in-sample network. The latter network perhaps benefited from seeing several interferograms for the same region in the training set (Marochov et al., 2021), and therefore, finds application in producing a time series of GLs for regions with a sufficient number of coherent interferograms."

We used the term 'out-of-sample' to refer to a dataset split in which the test samples are spatially apart from the training and validation samples, as used in Gourmelon et al. (2022). We admit that the deep learning community primarily uses 'out-of-sample' to refer to the test set. We apologize for the confusing terminology. Instead, we will refer to the 'in-sample' split as a 'temporal' split and the 'out-of-sample' split as a 'spatial' split.

Section 3.4 Training scheme explains the difference between the temporal and spatial dataset variants. Figure 1 is included as Fig. 6 in the revised manuscript.

(b) In Table 4, I think the feature subset should be one of these interferometric/non-interferometric feature combinations listed in Table 3? Why here is In-sample or out-of-sample? When you train two networks for in-sample and out-of-sample datasets, which interferometric/non-interferometric features combination did you use?

[Figure]

**Figure 1.** Spatial distribution of the AIS_cci lines into training (blue), validation (green) and test (red) sets (a) in-sample split (b) out-of-sample split. The in-sample data for (c) Abbot and (d) Amery Ice Shelf contain spatially overlapping but temporally separated training and test samples. In contrast, the out-of-sample data for the same ROIs contains only (e) test samples or (f) training samples. The in-sample data for the (g) Shackleton Ice Shelf contains training and validation samples, whereas the (h) Moscow University Ice Shelf samples contain only training samples. The out-of-sample data for the same ROIs (i) and (j) contain only training samples.

Thank you for noticing this discrepancy; we will change the column heading for Table 4 from "Features subset" to "Dataset split". As mentioned in Section 5.3, the two networks were trained on the rectangular features of the samples of the respective dataset splits:

Line 208: "To investigate HED's generalization capabilities, we used the network trained with rectangular features of the training samples from the in-sample dataset to delineate the test samples from the out-of-sample dataset (Fig. 7). We compared these delineations to those generated by HED, which was trained with the training samples of the out-of-sample dataset."

The heading in the first column of Table 4 has been corrected to "Dataset variant".

(c) From the paper itself, it seems you mainly used the in-sample training dataset to train the model and then evaluate the model performance on the in-sample and out-of-sample test sets, then what is the point of generating the out-of-sample training and validation sets?

Please refer to the response to comment 5b. We trained one of the networks with the training samples of the in-sample set and the other with the training samples of the out-of-sample dataset. The respective validation sets were also used to adjust the hyperparameters of the networks.

(d) Table 3 shows the numerical results of different networks, however, here it only shows results for one test set, is it an in-sample or out-of-sample test set?

We evaluated both networks on the out-of-sample split test set to make a sample-to-sample comparison. We mentioned this in Section 5.3 in the manuscript, cited in 5b. We will reiterate this point in Section 6.3.

We have included this in the revised manuscript, under Section 3.4, Line 172, "To evaluate our models' performance on realistic scenarios, we divided the samples in two ways, which we refer to as 'spatial' and 'temporal' variants (Fig. 6 a and b). The training, validation and test samples are spatially apart in the spatial split of the dataset, i.e., the training samples do not cover every glacier or ice stream. The training set of the temporal split contains a few samples from most ROIs, with the validation and test sets containing samples with different acquisition times ff the same ROI." We have repeated this in Section 4.3, line 268, "As stated in Section 3.4, the purpose of this experiment is to test our pipeline for an operational scenario where the network is used to generate long time series' of GLs for existing glaciers and ice streams in Antarctica."

6. I am not convinced by Section 6.1. The importance of the interferometric features can only be proved by comparing them with networks trained with non-interferometric features. However, here you only compare networks 1 & 2, which are both trained with interferometric features.

We intended to distinguish which among the two sets of interferometric features, rectangular or polar, are important for grounding line delineation: Line 193: "As mentioned in Section 3.2.1, we express the complex DInSAR interferograms as real and imaginary parts, i.e., the rectangular representation, as well as the pseudo coherence and phase components, i.e., the polar representation. We train two networks, one with rectangular interferometric features and the other with polar

interferometric features in their training features stack, to determine the optimal representation of the interferogram for GL delineation." We did train the network with just the non-interferometric features and decided to discard this experiment as the resulting delineations were too fragmented and unusable. We will mention this in the manuscript.

We have clarified this point in the revised manuscript, Line 234, "Although we performed an experiment in which HED was trained without interferometric features, we do not report its performance for two reasons. Firstly, the interferometric features are the only time-varying features in which the GL is reliably detectable. Secondly, the network trained with the only ice velocity and differential tide features produced no usable GLs. The model performance improved drastically with the addition of the DEM. Later in this section, we elaborate on the implications of using the DEM."

7. Section 6.2 the importance of DEM (Line 270 and Figure 10):

   – Please include a detailed zoomed-in map of the interferogram inside the blue box. It seems the interferogram phase inside the blue box is decorrelated, so I won't be surprised that the network cannot map the correct GL. Also, only giving one example with a small spatial extent is not representative.

     We will provide a zoomed-in inset of the interferogram inside the blue box. Indeed, the interferogram in this region is partially decorrelated, making it likely that the network relies more on the DEM. While we agree that the manual GL in this region may not be 100% accurate, we do not believe that the real grounding line coincides with the elevation drop shown in the DEM. Ideally, we expect the network to not make any delineations at all for such cases. We will provide other examples as well.

   – Have you checked the elevation change in Cabinet Inlet, is it a region undergoing significant elevation changes? If elevation is stable, I don't think you can attribute the wrong GLs to different DEM stacks

     Thank you for raising this valid concern. We will have a look to see if it is stable. You are right; if the region is stable, the break in slope proxy derived from the DEM could be close to the interferometric grounding line. However, the more significant point we were trying to make is that our DEM feature does not vary temporally; therefore, one must be cautious when training the network with the DEM.

     We have corrected this in the revised manuscript in Section 4.2, Line 253, " Apart from a few isolated data samples, we did not find any conclusive evidence that this was the case for our model. However, a few models developed for automatic delineation of calving fronts were shown to overfit to the DEM (Heidler et al., 2022; Loebel et al., 2022). While we understand that grounding line migration is not as dynamic as glacier calving, we advise future users to verify that the ROI is relatively stable so that adding the DEM does not misguide the network." The figure for Cabinet Inlet has been removed.

   – How to achieve the balance of including DEM to avoid over-reliance?
     We have yet to explore this aspect in our study, so, unfortunately, we cannot suggest a strategy to overcome this problem. It is of scientific interest and would be worth investigating in a follow-up study.

8. Section 6.3:

(a) As mentioned above, I don't understand why compile two different in-sample/out-of-sample sets and train two networks. If you combine the in-sample and out-of-sample sets into one dataset, won't this greatly increase the training samples and improve the model performance?

Please refer to the response to comment 5a.

(b) You evaluate the in-sample trained model performance on the unseen Ross Ice Shelf interferograms by using Figure 12, however the discussion on the prediction quality is limited. Most GZ regions in Ross Ice Shelf are stable, I would like to see a distance deviation map between the AIS_cci GL and the network-generated GLs in Ross Ice Shelf to demonstrate the performance. If there are large deviations, please consider explaining 1) what are causing the large deviation? 2) which dataset is correct? 3) how can you further improve these results?

Given the stability of the glaciers in the region, we agree that this is an 'easy' ROI. Still, the network was never trained with any interferograms from the Ross Ice Shelf. Fig. 12 was used only to show the spatial transfer capability of the network and gauge the quality of the delineations from a purely visual perspective, as these interferograms were never manually delineated. The black lines in Fig. 12 (a) correspond to a different set of double differences, for which we have manual delineations.

Line 291: "Despite never having seen the interferograms during the training or validation stages, the network delineated the landward-most fringe and largely avoided delineating the decorrelated fringes of the Crary Ice Rise and Nimrod Glacier interferograms in Fig. 12c, d. The loose fringes in the interferograms of the Dickey Glacier and Nursery Glacier were not delineated. The network complemented existing manual delineations, reducing significant gaps and leading to a more complete grounding line in this area."

Nevertheless, we agree that a quantitative performance assessment is more convincing than a qualitative one. We will add a figure showing the deviation between the existing manual and network delineations and augment the discussions as suggested.

This figure (now Fig. 10 in the revised manuscript) remains unchanged, except for inverting the colors of the HED and AIS_cci GLs. As stated in our initial response, the HED delineations have been made for previously unseen interferograms, which have not been manually delineated either. Since the interferograms are highly coherent, we believe a visual inspection can convince readers of HED's performance.

(c) In addition, I am curious to know what new GL information you can provide by using your approach. What is the implication of using your model in mapping the GLs and improving our understanding of the GL migrations?

We have partially addressed this in the conclusion: Line 314: "We also demonstrated the ability of HED to delineate interferograms of previously unseen regions without retraining the network, which enables the timely delineation of new interferograms without manual intervention. Our delineation pipeline is independent of the source of the SAR scenes and, therefore, can be used to delineate coherent DInSAR interferograms provided by any spaceborne SAR mission."

Obtaining a dense time series of GLs enables migration analysis. We will add this point to the conclusion.

This sentence has been added to the revised manuscript in Line 334, "Therefore, the network could be used to generate a large dataset to enable GL migration analysis at various timescales."

9. Figures:

(a) Please consider labeling all the subplots in each figure, and adding a subplot to show the ROI location in Antarctica.

(b) Figure 8, it is impossible to visually compare the differences between GL predictions from these two networks given the current presentation format. I suggest plotting the spatial deviations between the network predictions so we can directly see where and how much these two are deviating from each other. Again, there are multiple ways to visualize this difference.

(c) Same problem with Figure 9:

   – Cross-referencing the three inset figures by just coloring the subplot figure frames is not helpful.

   – On Larsen C Ice Shelf, it is impossible to see the details of network-generated GLs inside the green box in the first subplot.

   – The plotting extent cut out the GLs in Totten main glacier stream, you need to expand the spatial extent.

   – Why not also plot the three inset boxes in the second column?

   – In the final column, you present the zoomed-in interferograms and show the manual GLs, why not plot the network-generated GLs from these two different networks so we will know the different performances of these two networks in Totten?

(d) Figure 11:

   – It's difficult to compare these two outputs without putting them in the same figure or providing a distance deviation map.

   – You have done an Antarctica-scale evaluation, why not include a comparison map for the whole ice sheet?

We will improve the visibility of our figures, as suggested. We will include an Antarctic-wide heat map of the deviations between AIS_cci and network delineations.

The figures have been improved as per the reviewers' suggestions. Figure 13. shows an Antarctic-wide overview of the PoLiS distances in the form of a boxplot.

**Technical Comments**

Line 15: provide the mass change uncertainty for both ice sheets.

The interval of mass change uncertainty was mentioned in square brackets, as provided in Fox-Kemper et al. (2021). Line 15: "...the amount of ice mass loss from the Antarctic Ice Sheet (AIS) (4890 [4140–5640] Gt) and Greenland Ice Sheet (2670

[1800-3540] Gt)"

We will change these to ± standard deviations to improve readability.

This change is included in Line 15, "...ice mass loss from the Antarctic Ice Sheet (AIS) (2671 ± 530 Gt) and Greenland Ice Sheet (4892 ± 457 Gt)...".

Line 25-50: these three paragraphs need restructuring:

- Grounding line itself is a subglacial feature, please elaborate why detecting these two features is challenging and why different (surface) features can be used as proxies for the grounding line.

  We have mentioned the difficulties in detecting the grounding line in the manuscript: Line 26: "There are two main challenges when detecting the grounding line: its sub-glacial location and the short-term migration due to the tidal flexure of the ice shelf." We agree with you that we use "surface" observations, such as the tidal deformation of the ice, to detect a subsurface proxy for the grounding line, i.e., the hinge line. We will clarify this in the revised manuscript.

  The paragraph has been modified from Line 26, "There are two main challenges when detecting the grounding line: its sub-glacial location and the short-term migration due to the tidal flexure of the ice shelf. The former problem is addressed by considering other features as proxies for the true GL (Brunt et al., 2011). These features are illustrated in Fig. 1a. A few works have investigated the latter issue (Milillo et al., 2019; Chen et al., 2023; Freer et al., 2023, revealing the grounding line's highly localized and heterogeneous movement with ocean tides. We will not delve into details about this as our work focuses on solving the problem of automatically detecting the grounding line from satellite data."

- You first cite Brunt et al., 2011 to say that existing methods detect grounding line proxies, then talk about using ice-penetrating radar in detecting true grounding line G which is a subglacial feature. The logic here is problematic.

  Thank you for pointing out the erroneous logic. We will restructure these sentences and bring them into the correct order.

  The restructured paragraph now reads (Line 32), "Despite being located underneath hundreds of meters of ice, some in-situ methods such as terrestrial (Jacobel et al., 1994; Catania et al., 2010; MacGREGOR et al., 2011) and airborne (Uratsuka et al., 1996) ice-penetrating radar have been used to derive the true grounding line G. Other in-situ methods such as surface slope measurements using tiltmeters (Stephenson and Doake, 1979; Stephenson, 1984; Smith, 1991), static Global Positioning Systems (GPS) (Riedel et al., 1999) and kinematic GPS (Vaughan, 1994) measure the hinge line location F. Although the GLs derived from these methods have limited spatial and temporal resolution, they serve to validate GLs derived from remote sensing methods."

Line 51: it is 'grounding line' not 'grounded line'.

Thank you for catching this typo. We will correct this.

The typo has been corrected.

Line 54: where is this research 'Ramanath Tarekere, 2022' published?

Thank you for bringing this to our attention. We will correct this reference.

The reference has been corrected.

Line 63-64: ICESat laser altimetry has also been used in generating grounding zone products manually by Fricker et al. (2006, 2009) and Brunt et al. (2010, 2011).

We will cite these references in the Introduction.

The citations have been included, Line 40, "Elevation profiles derived from radar altimetry (Dawson and Bamber,2017, 2020; Hogg et al., 2018) and laser altimetry (Fricker and Padman, 2006; Fricker et al., 2009; Brunt et al., 2010, 2011; Li et al., 2020) capture pointwise measurements of $F$, the seaward limit of the tidal flexure, $H$, and the break in slope, $I_b$ (Fig. 1a)."

Line 65: I see what you are trying to say here – emphasizing DL method does not need manual intervention compared to other methods. However, I find it a bit confusing to follow the logic. Having read the first sentence, I would expect to know the research progress in using DL methods in detecting GZ, but here you directly dive into model inversion and ICESat-2 methods.

We mentioned both DL and classical techniques: Line 65: "Nonetheless, recent research endeavours have explored deep learning (DL) and classical model inversion techniques for automatic GL detection.". Nevertheless, this gives the impression that we will explain the DL methods first. We will rewrite this sentence and explain the classical methods first and the DL methods after that.

This sentence no longer exists in the revised manuscript.

Line 74-79: In addition to laser altimetry, there are several studies that have used CryoSat-2 radar altimetry in mapping GZ automatically, such as Dawson and Bamber (2017, 2020), and Hogg et al. (2018).

We have indeed missed to cite these important references in our manuscript. Thank you for bringing this to our attention; we will include them in the revised manuscript.

We have added these citations, please refer to the above response.

Line 138: the pyTMD should be cited as Sutterley et al. (2017). Check https://pytmd.readthedocs.io/en/latest/getting_started/Citations.html

Thank you, we will cite the correct reference.

We have corrected this reference.

Line 175: how did you determine 0.8 as the threshold?

We performed a grid search over the thresholds $[0.5, 0.9] \in \mathbf{R}$ and chose the threshold which resulted in the smallest PoLiS distance between the AIS_cci GLs and the network delineations. However, the threshold value will likely change as we now use the delineations from an ensemble. We will mention the grid search method in the manuscript.

We have mentioned this protocol in Section 3.5, Line 183, "The ODS F1 score is computed considering all dataset samples after converting the corresponding predictions to a binary map at a threshold value that yields the highest score. For our dataset, this value is 0.8."

Line 278-279: can you explain more about this claim? Given the current evidence in this section, I don't follow how you can claim that HED relies more on the rectangular interferometric features or DEM than the non-interferometric features.

Thank you for pointing out this statement. Indeed, we cannot claim that the network relies more on the rectangular features or the DEM when non-interferometric features are present. The non-interferometric features have either a confounding or insignificant impact on the resulting delineations. We will change this sentence to the one stated above.

This is now clearer in Line 262, "When combined with the DEM, the other non-interferometric features seem to have an insignificant impact on the resulting delineations (networks 7, 9 and Fig. B3).

Figure 2: It should be differential tidal amplitude

Thank you for pointing out this inconsistency; we will correct the subplot heading.

Figure 2 has been corrected.

Figure 5: I am confused about this figure:

- The subplot in the second row of the second column 'Resample Inputs', what are these two red boxes? Are these two different sampling locations that correspond to two different interferogram subsets in the third column? Also, what is the meaning of those three dots?

- I suggest replotting this figure to make it as clear as possible.

The red squares show two tiles/patches of one interferogram (the ROI being Amery Ice Shelf). The column 'Interferogram and Manual delineation tiles' shows the zoomed-in tiles highlighted by the red squares in the column 'Resampled Inputs'. The three dots indicate that the same process follows for each interferogram's tiles. We will include this explanation in the figure caption.

Figure 5 has been changed as per suggestions.

**References**

Fox-Kemper, B., Hewitt, H. T., Xiao, C., Aðalgeirsdóttir, G., Drijfhout, S. S., Edwards, T. L., Golledge, N., Hemer, M., Kopp, R., Krinner, G., Mix, A., Notz, D., Nowicki, S., Nurhati, I., Ruiz, L., Sallée, J.-B., Slangen, A., and Yu, Y.: Ocean, Cryosphere and Sea Level Change, Climate Change 2021: The Physical Science Basis. Contribution of Working Group I to the Sixth Assessment Report of the Intergovernmental Panel on Climate Change, pp. 1211—-1362, 2021.

Gawlikowski, J., Tassi, C. R. N., Ali, M., Lee, J., Humt, M., Feng, J., Kruspe, A., Triebel, R., Jung, P., Roscher, R., et al.: A survey of uncertainty in deep neural networks, Artificial Intelligence Review, 56, 1513–1589, https://doi.org/10.1007/s10462-023-10562-9, 2023.

Gourmelon, N., Seehaus, T., Braun, M., Maier, A., and Christlein, V.: Calving fronts and where to find them: a benchmark dataset and methodology for automatic glacier calving front extraction from synthetic aperture radar imagery, Earth System Science Data, 14, 4287–4313, 2022.

Guidicelli, M., Huss, M., Gabella, M., and Salzmann, N.: Spatio-temporal reconstruction of winter glacier mass balance in the Alps, Scandinavia, Central Asia and western Canada (1981–2019) using climate reanalyses and machine learning, The Cryosphere, 17, 977–1002, https://doi.org/10.5194/tc-17-977-2023, 2023.

Herrmann, O., Gourmelon, N., Seehaus, T., Maier, A., Fürst, J. J., Braun, M. H., and Christlein, V.: Out-of-the-box calving-front detection method using deep learning, The Cryosphere, 17, 4957–4977, https://doi.org/10.5194/tc-17-4957-2023, 2023.

Lakshminarayanan, B., Pritzel, A., and Blundell, C.: Simple and Scalable Predictive Uncertainty Estimation using Deep Ensembles, in: Advances in Neural Information Processing Systems, edited by Guyon, I., Luxburg, U. V., Bengio, S., Wallach, H., Fergus, R., Vishwanathan, S., and Garnett, R., vol. 30, Curran Associates, Inc., https://proceedings.neurips.cc/paper_files/paper/2017/file/9ef2ed4b7fd2c810847ffa5fa85bce38-Paper.pdf, 2017.

Mohajerani, Y., Jeong, S., Scheuchl, B., Velicogna, I., Rignot, E., and Milillo, P.: Automatic delineation of glacier grounding lines in differential interferometric synthetic-aperture radar data using deep learning, Scientific reports, 11, 1–10, 2021.

Ramanath Tarekere, S.: Mapping the grounding line of Antarctica in SAR interferograms with machine learning techniques, Master's thesis, Technische Universität München, https://elib.dlr.de/189234/, 2022.

Valdenegro-Toro, M.: Deep sub-ensembles for fast uncertainty estimation in image classification, arXiv preprint, https://doi.org/10.48550/arXiv.1910.08168, 2019.

---

## Referee Report (RR1)

**Review Comments for "Deep Learning Based Automatic Grounding Line Delineation in DInSAR Interferograms"**

Thank you for addressing my comments in the first round of revision. While I appreciate the improvements made, I believe this paper still needs further clarification and improvement before I can recommend this manuscript for publication in The Cryosphere. The current version has some ambiguity regarding the main conclusions about which features are most effective for the neural network in mapping grounding lines. Additionally, the results sections provide limited insights into the new grounding line information generated by your proposed method, making it challenging to fully understand its utility.

**General Comments**

The narrative needs improvements to better highlight key conclusions about feature selection and network performance. It is unclear which features and network configuration are recommended for actually mapping the grounding line from DInSAR.

The results need to expand on how the new grounding line data derived from your method adds value or compares to existing datasets.

**Specific Comments:**

Line 6: Consider rephrasing to "delineating the grounding line from DInSAR interferograms" instead of "delineating DInSAR interferograms."

Line 9: You mention assessing the contribution of non-interferometric features. However, the conclusions from this assessment are unclear. What are the most important features identified, and which network configuration performs best? The Results sections (4.1 and 4.2) need clearer writing to highlight these conclusions.

Line 11: Can you explain why the mean distance has such a high uncertainty?

Line 15: The statement "while there is little doubt in the amount..." is not entirely accurate. Different satellite-based mass balance measurements produce different values for mass change. What is consistent is the trend. See https://www.nature.com/articles/s41586-018-0179-y

Line 26: Short-term grounding line migration isn't a challenge for mapping using satellite images, as satellite images represent snapshots.

Line 30: If tide-induced grounding line variability is irrelevant to this paper, consider removing it from the introduction.

Line 34: Can surface slope measurements from tiltmeters actually identify Point F? I think it's break in slope?

Line 46: The sentence "The above-stated methods and others... detailed in Friedl et al., 2020" could be better integrated into the paragraph for conciseness.

Lines 59–66: The details on Parizzi (2020) seem excessive for this context especially its method is not super relevant to your approach. Please revise the description to maintain focus on your study.

Line 74: please provide details about your earlier works (Ramanath Tarekere, 2022 and 2023). Explain what you have done in your initial works, how the methods differ in this study and why further improvements to Mohajerani (2021) are necessary.

Line 83: Include details on where the AIS_cci GLL product can be accessed?

Line 117: Why only incorporate northing and easting components? Please justify this decision.

Line 162: The example in Figure 5 does not clearly illustrate the use of the median filter for postprocessing in eliminating spurious branches'. Consider using a more illustrative example.

Line 175: What is meant by "most ROIs"? Please specify.

Line 177: Figure 6h fits the description, but so do Figures 6g, 6i, and 6j. Clarify this point.

Line 239: Could you provide evidence to support the statements here?

Line 240: Instead of saying, "Later in the section, we elaborate...," consider integrating this explanation directly into the paragraph. This will make the section flow better.

Line 248: Have you considered discussing Network 3 here as well?

Line 250: using a break in slope as a proxy for Point F can be problematic, especially for fast-flowing glaciers. If DInSAR data is unreliable in these regions, the break-in-slope method is likely to have similar limitations. Additionally, on Line 257, you suggest that users verify the stability of the ROI to avoid errors when adding the DEM. This seems to contradict your earlier statement about the benefits of using the break in slope in fast-flowing areas for detecting the grounding line when DInSAR interferograms are unavailable.

Sections 4.1 & 4.2:

- In Section 4.1, I was curious why not train a network that combines both the rectangular representations and the polar representations? Additionally, I noticed in Table 3 that networks 3–9 were all generated using the real and imaginary components. Could you clarify why phase and pseudo-coherence weren't used instead?
- You've talked about the contributions of interferometric and non-interferometric components in mapping the grounding line. However, one critical piece of information is missing: which network do you recommend using? This isn't clear in these sections but is only briefly mentioned in the conclusion.

Line 266: You mention using Network 1, which focuses on rectangular features. However, according to Table 3, Network 3 has the best performance. Could you explain why Network 3 wasn't used instead?

Lines 273-275: Could you provide evidence to support this? Should we refer to Figure 9 here?

Lines 280-286: Did you apply this method to the entire Antarctic Ice Sheet? If so, how does your new grounding line (GL) data compare to other publicly available GL products? Expanding this section with such comparisons would be very helpful.

**Figures:**

Please review all the figures to ensure they are easy to read and visually clear for the readers.

Figure 1: Could you add a geolocation map to Fig. 1b? This would help provide more context for the figure.

Figure 2: It might be clearer to separate the features into two columns: one for interferometric features and another for non-interferometric features. The current color choices and line widths make it challenging to easily distinguish between these categories. Additionally, consider labeling the interferometric and non-interferometric features directly in the figure to improve clarity.

Figure 4: Is this network architecture identical to the one in Xie and Tu (2015)? If not, could you clarify whether it is based on that work?

Figure 5: The subtitles above the figure panels are quite small and hard to read when the paper is printed on A4 paper. It would be helpful to increase the font size. The same issue applies to Figure 6.

Figure 7: The shapes of the inset polygons in columns 1 and 3 don't match the shapes of the zoomed-in maps in columns 2 and 4. The insets are rectangular, while the zoomed-in maps are square. Please adjust this for consistency. Additionally, adding titles or annotations directly within the figures to distinguish between different networks would make it more straightforward for readers. Referring to "columns 1 & 3" or "columns 2 & 4" in the text can be unclear.

Figures 8 & 9: These figures have the same issue as Figure 7 regarding inset shapes and labeling. Please consider making similar adjustments.

Figure 10: There is an error in the subplot labels for (a) and (c). Crary Ice Rise should be labeled as (a), not (c).